# Energy-saving hydrogen production by chlorine-free hybrid seawater splitting coupling hydrazine degradation

Fu Sun[1], Jingshan Qin[2], Zhiyu Wang [1✉], Mengzhou Yu[3], Xianhong Wu[1], Xiaoming Sun[2✉] & Jieshan Qiu [1,4✉]

Seawater electrolysis represents a potential solution to grid-scale production of carbon-neutral hydrogen energy without reliance on freshwater. However, it is challenged by high energy costs and detrimental chlorine chemistry in complex chemical environments. Here we demonstrate chlorine-free hydrogen production by hybrid seawater splitting coupling hydrazine degradation. It yields hydrogen at a rate of 9.2 mol h$^{-1}$ g$_{cat}$$^{-1}$ on NiCo/MXene-based electrodes with a low electricity expense of 2.75 kWh per m$^3$ H$_2$ at 500 mA cm$^{-2}$ and 48% lower energy equivalent input relative to commercial alkaline water electrolysis. Chlorine electrochemistry is avoided by low cell voltages without anode protection regardless Cl$^-$ crossover. This electrolyzer meanwhile enables fast hydrazine degradation to ~3 ppb residual. Self-powered hybrid seawater electrolysis is realized by integrating low-voltage direct hydrazine fuel cells or solar cells. These findings enable further opportunities for efficient conversion of ocean resources to hydrogen fuel while removing harmful pollutants.

[1] State Key Lab of Fine Chemicals, Liaoning Key Lab for Energy Materials and Chemical Engineering, PSU-DUT Joint Center for Energy Research, Dalian University of Technology, Dalian, China. [2] College of Chemistry, Beijing University of Chemical Technology, Beijing, China. [3] State Key Laboratory of Space Power-Sources Technology, Shanghai Institute of Space Power-Sources, Shanghai, China. [4] College of Chemical Engineering, Beijing University of Chemical Technology, Beijing, China. ✉email: zywang@dlut.edu.cn; sunxm@mail.buct.edu.cn; jqiu@dlut.edu.cn

Hydrogen ($H_2$) represents the ultimate choice of sustainable and secure energy due to its superior energy density of 142.351 MJ kg$^{-1}$ and zero-pollution emission[1]. According to the International Renewable Energy Agency, the market value of hydrogen feedstock would boost to $ 155 billion by 2022 as the beginning of the global hydrogen economics[2]. Water electrolysis excels the traditional petrochemical techniques in terms of processing efficiency, renewables compatibility, and carbon neutrality for yielding high-purity hydrogen[3,4]. But this technology produces only 4% of hydrogen in the market due to the unaffordable cost (>$ 4 kg$^{-1}$) of electricity consumption for overcoming the high potential of overall water splitting (OWS) reaction[5]. Another rarely noticed but critical concern is the demand on large quantities of high-purity water feeds for water electrolysis. It would become a bottleneck to the deployment of this technology in the arid, on and off-shore areas. The ocean provides 96.5% of the planet's water reserve, providing infinite hydrogen sources without heavy strain on the global freshwater resource[6-8]. However, the electrolysis of seawater with complex ionic chemistry faces extra challenges in dealing with the interference of side reactions, ionic poison, and corrosion on cell performance. A notorious problem is the chlorine electro-oxidation reactions (ClOR) and their competition with oxygen evolution reaction (OER) on the anode. This reaction releases toxic and corrosive chlorine species (e.g., $Cl_2$, $ClO^-$), which induces anode dissolution and environmental hazards to reduce the electrolysis efficiency and sustainability[9-12]. The ClOR can be suppressed by limiting the OER overpotential below 0.48 V under alkaline conditions (Fig. 1a). However, minimizing the polarization overpotential requires conducting the electrolysis at the current densities (<200 mA cm$^{-2}$) far lower than the industrial criteria (>500–1000 mA cm$^{-2}$)[7,13]. Applying the cation-selective layer or chlorine-free anolyte is effective in protecting the anode from chlorine corrosion at industrially required current densities[9,13]. Nevertheless, it is still hard to eliminate the chlorine crossover and corrosion for long-term seawater electrolysis, and the process suffers high cell voltages (>1.7–2.4 V) and electricity consumption. So far, the development of chlorine-free yet energy-saving seawater electrolysis technology still remains challenging for cost-effective and sustainable hydrogen production[7,14].

For commercial alkaline water electrolyzers, the basic electricity demand is 4.3–5.73 kWh for yielding 1 m$^3$ of $H_2$ at the cell voltages of 1.8 − 2.4 V and practical current level of 300−500 mA cm$^{-2}$[15-17]. Such a high energy consumption fundamentally stems from the OER with large thermodynamic potential (1.23 V vs.

RHE) and slow multiple proton-coupled electron-transfer kinetics[18,19]. This bottleneck is difficultly broken as long as the sluggish OER is involved in OWS. Replacing the OER by thermodynamically more favorable electro-oxidation reactions offers a ground-breaking strategy for energy-saving hydrogen production while adding extra functionalities like electrosynthesis[20-24]. Among available options, the hydrazine oxidation reaction (HzOR, $N_2H_4 + 4OH^- \rightarrow N_2 + 4H_2O + 4e^-$, −0.33 V vs. RHE) holds great potential for yielding hydrogen at much lower voltages than OER with zero pollution emission[25-28]. For seawater electrolysis, the oxidation potential of HzOR is far lower than that of ClOR by 2.05 V (Fig. 1a). This advantage provides the extra benefit in avoiding the notorious problems of chlorine chemistry without limiting the electrolysis current and hydrogen yielding efficiency. On the other hand, hydrazine has been widely used as the deoxidant in the feedwater of power plants, high-energy fuel, and raw materials in the industry. The fabrication, utilization, and disposal of this highly toxic material may severely risk human health and the ecosystem[29]. Efficient technologies are thus highly desired to degrade the hydrazine in surface water to a trace residual (e.g., 10 ppb in drinking water set by U.S. Environmental Protection Agency, EPA)[30]. Electrocatalytic HzOR offers a promising way for fast removal of hydrazine from industrial sewage without using extra oxidants (e.g., Fenton's reagent) or complex separation[31]. Integrating this technique into the electrolysis of costless seawater is anticipated to bring great benefits in not only environmental sustainability but also the cost-effectiveness of hydrogen production.

Herein, we propose to realize energy-saving yet chlorine-free seawater electrolysis for efficient hydrogen production by a hybrid seawater splitting strategy. This chemistry consumes the seawater on the cathode to generate $H_2$ by hydrogen evolution reaction (HER); while the crossover of released $OH^-$ to the anode side supply the hydrazine degradation to harmless $N_2$ and water with reduced salinity (Fig. 1b). Beyond the state-of-the-art seawater electrolysis, it enables hydrogen production at ultralow cell voltages but large current densities without chlorine hazards and limiting hydrogen-yielding efficiency. The hybrid seawater electrolyzer (HSE) using NiCo/MXene-based superaerophobic-hydrophilic and hydrazine-friendly electrodes requires a dramatically lower electricity expense of 2.75 kWh/m$^3$ $H_2$ than alkaline seawater electrolyzer (ASE) at industrial-scale current densities. This electrolyzer simultaneously allows fast hydrazine degradation to a rather lower residual while harvesting water with reduced salinity from seawater. On this basis, self-powered

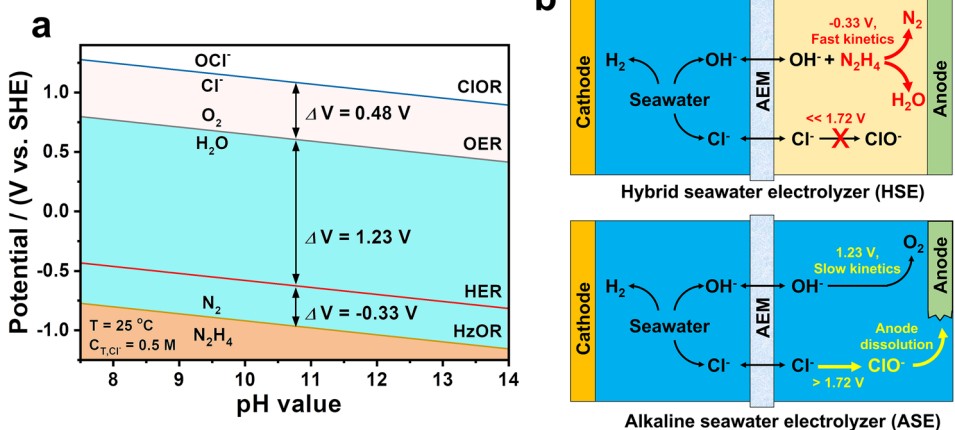

**Fig. 1 Schematic illustration of the merits of hybrid seawater splitting for energy-saving and sustainable hydrogen production. a** The Pourbaix diagram of HzOR, HER, OER, and ClOR in artificial seawater with 0.5 M Cl$^-$ in pH 7–14. **b** The merits of HSE over ASE for energy-saving and chlorine-free hydrogen production.

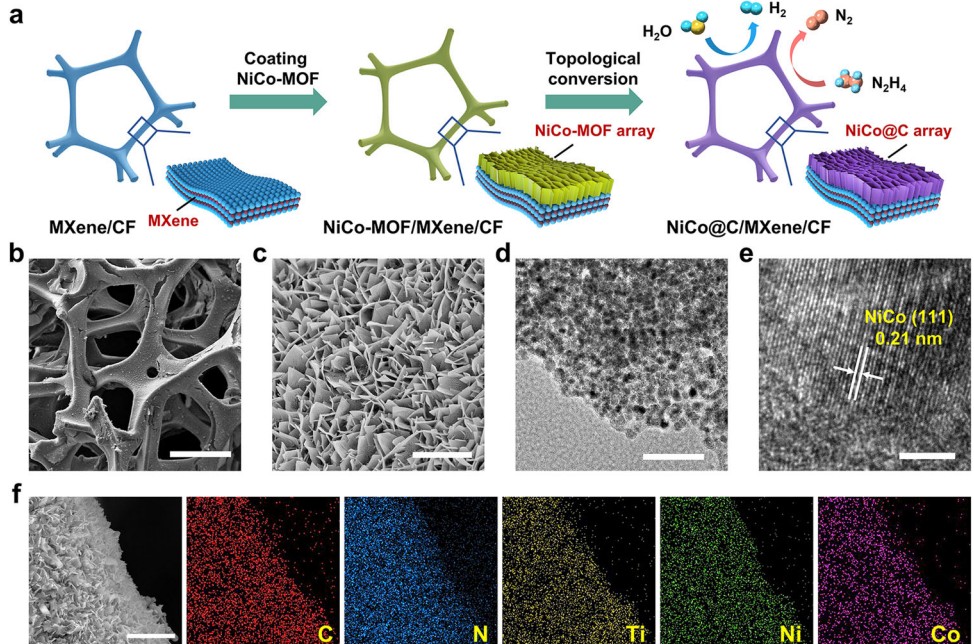

**Fig. 2 Characterizations of NiCo@C/MXene/CF. a** Schematic illustration of the synthetic strategy of NiCo@C/MXene/CF. **b** SEM image showing macroporous scaffold of this electrode. Scale bar, 200 μm. **c** SEM image of mesoporous networks of NiCo@C nanosheets on electrode surface. Scale bar, 3 μm. **d** TEM image of a NiCo@C nanosheet. Scale bar, 100 nm. **e** HRTEM image of NiCo nanocrystallite on the nanosheet. Scale bar, 3 nm. **f** Elemental mapping showing the uniform distribution of C, N, Ti, Ni, and Co elements in this electrode. Scale bar, 5 μm.

seawater electrolysis can be further realized by integrating the HSE into solar or hydrazine fuel cells for better cost-effectiveness and sustainability.

## Results

**Synthesis and characterization of NiCo/MXene-based electrode**. A NiCo/MXene-based electrode is designed to optimize the gas-releasing capability and water/hydrazine compatibility for propelling hybrid seawater splitting. It is fabricated by assembling NiCo-MOF nanosheets on MXene-wrapped Cu foam (MXene/ CF, Supplementary Fig. 1), followed by annealing in $NH_3$ (denoted as NiCo@C/MXene/CF, Fig. 2a). Scanning electron microscopy (SEM) and Atomic force microscopy (AFM) reveal the formation of a mesoporous array of NiCo-decorated carbon nanosheets (NiCo@C) with an average size of 400–800 nm and a thickness below 50 nm on MXene/CF (Fig. 2b, c and Supplementary Fig. 2). Such nanoarray-based rough surface can largely weaken the gas adhesion on discontinuous solid-liquid-gas triple-phase contact dots, thereby enabling a superaerophobic property[32,33]. While the MXene layer with abundant -OH and -O groups may effectively attract the water and hydrazine molecules via hydrogen bonding interaction[34]. Coupling them into 3D configuration yields an electrocatalytic electrode with superaerophobic-hydrophilic and hydrazine-friendly interface, large gas transport channels, high active surface area and superb conductivity for promoting hybrid seawater electrolysis. Transmission electron microscopy (TEM) and X-ray diffraction (XRD) reveal that the NiCo@C consists of numerous single-crystal NiCo alloy nanoparticles (<10–20 nm) embedded in the amorphous carbon matrix (Fig. 2d, e and Supplementary Fig. 3)[35]. The face-centered cubic (fcc) structure of NiCo alloy in NiCo@C/MXene is identified by a rather similar XRD pattern with fcc Ni in Ni@C/ MXene and fcc Co in Co@C/MXene. Coordination states of the metal atoms in NiCo@C are further investigated by X-ray absorption fine structure (XAFS). The NiCo@C, Ni@C and

Co@C nanosheets are peeled off from the CF to minimize the influence of CF on the analysis. The K-edge X-ray absorption near-edge structure (XANES) spectra of Ni and Co in NiCo@C are close to that of Ni@C, Co@C and metal foil reference, sug-gesting a metallic state of these elements (Supplementary Fig. 4a, b). Curve fitting of Fourier-transformed extended X-ray absorp-tion fine structure (FT-EXAFS) spectra reveal the change of coordination number of Ni from 8.3 in Ni@C to 9.6 in NiCo@C while the value of Co increase from 8.4 in Co@C to 9.0 in NiCo@C (Supplementary Fig. 4c, d). This phenomenon indicates the formation of NiCo alloy instead of their mixture. The pre-sence of Ni-Ni bonds in NiCo alloy is identified by a similar metal bond length in NiCo@C (2.64 Å) and Ni@C (2.63 Å). The alloying of Co with Ni with a smaller atomic size induces a shorter metal bond length of 2.56 Å with respect to Co-Co bonds in Co@C (2.63 Å), implying the atomic dispersion of Co atoms in Ni lattice in NiCo alloy. Tiny oxidization states (Ni-O, Co-O) appear for all the samples and metal foil references due to inevitable surface oxidation during XAFS analysis in air. No nickel or cobalt nitrides are formed by annealing NiCo-MOF at a relatively low temperature, which is consistent with the XRD result. Elemental mapping and X-ray photoelectron spectroscopy (XPS) visualize the presence and uniform distribution of C, N, Ti, Ni, and Co elements on NiCo@C/MXene/CF (Fig. 2f and Sup-plementary Fig. 5a). The Ni exists as a metallic state with a strong $2p_{3/2}/2p_{1/2}$ doublet at 852.8/870.18 eV in Ni $2p$ spectrum (Sup-plementary Fig. 5b)[36]. The Co has a mixed metallic and $Co^{2+}$ state, identified by the doublets at 778.5/793.3 and 780.7/796.3 eV in Co $2p$ spectrum, respectively (Supplementary Fig. 5c)[37]. The Ti $2p$ spectrum validates the presence of $Ti_3C_2T_x$ MXene by the signals from Ti-C lattice with $Ti^{2+}$ and $Ti^{3+}$ species that grafting the surficial groups via Ti-O and Ti-F bonds (Supplementary Fig. 5d, e)[19]. Strong $Ti^{3+}$ signal may be a result of the reaction between oxygen-containing groups on MXene surface and Ti atoms connected to them during annealing. To achieve the best HzOR and HER activity, the NiCo content is optimized to 84.7

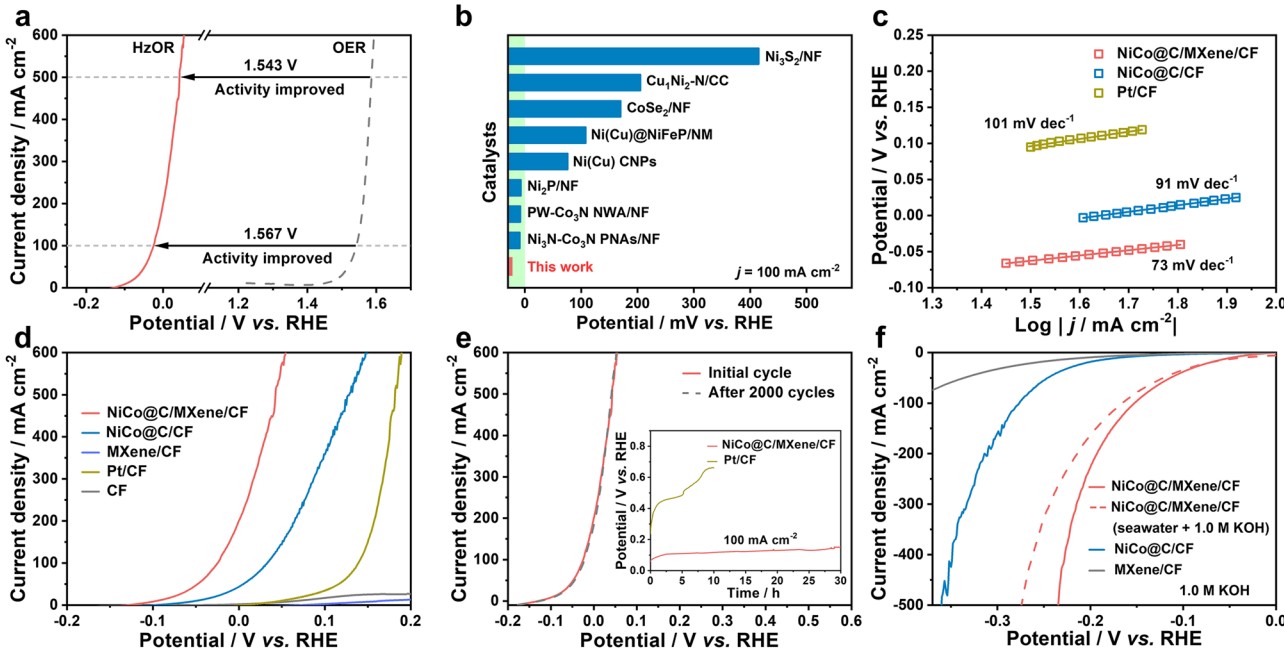

**Fig. 3 Half-cell HzOR and HER performance of NiCo@C/MXene/CF. a** A comparison between HzOR and OER in 1.0 M KOH in anode potential. **b** A comparison between NiCo@C/MXene/CF and reported 3D electrodes in HzOR activity. **c** Tafel plots of NiCo@C/MXene/CF, NiCo@C/CF and Pt/CF for HzOR. **d** The LSVs of NiCo@C/MXene/CF and controlled catalysts including NiCo@C/CF, MXene/CF, Pt/CF and CF for HzOR. **e** The LSVs initially and after 2000 sweeps for HzOR. The insert is the chronopotentiometric curves of NiCo@C/MXene/CF and Pt/CF for HzOR at a current density of 100 mA cm$^{-2}$. All HzOR tests are conducted in 1.0 M KOH with 0.5 M N$_2$H$_4$ at a scan rate of 10 mV s$^{-1}$. **f** The LSVs for HER in 1.0 M KOH or seawater (pH 13.8) at a scan rate of 10 mV s$^{-1}$. The HER activity of controlled catalysts including NiCo@C/CF and MXene/CF is also compared under identical conditions.

wt.% in NiCo@C with a Ni: Co ratio of 2.7: 1 (Supplementary Fig. 6, 7). The average NiCo@C loading on MXene/CF is around 1.0 mg cm$^{-2}$.

**Half-cell HzOR performance of NiCo/MXene-based electrode.** The HzOR activity of NiCo@C/MXene/CF is evaluated in 1.0 M KOH with various hydrazine concentrations. All the potentials are against the RHE unless otherwise specialized. This electrode exhibits a rapid increase in activity with hydrazine concentration increasing until 0.5 M (Supplementary Fig. 8a). In 1.0 M KOH with 0.5 M hydrazine, achieving high current densities of 100 and 500 mA cm$^{-2}$ only requires ultralow potentials of −25 and 43 mV, respectively (Fig. 3a and Supplementary Fig. 9a). In sharp contrast, the sluggish OER needs 37–62 times higher potential (1.542–1.586 V) for reaching the same current level in 1.0 M KOH. Compared to reported 3D electrodes, the NiCo@C/MXene/ CF still exhibits far superior HzOR activity under similar conditions (Fig. 3b and Supplementary Table 1). Such a dramatic reduction in anode potential is rather beneficial to hydrogen production at a low cost of electricity. With scan rate increasing from 5 to 100 mV s$^{-1}$, the HzOR proceeds with a negligible shift of the polarization curves, indicating fast kinetics across the electrode-electrolyte-gas triple-phase interface on NiCo@C/ MXene/CF (Supplementary Fig. 8b). Fast charge transfer kinetics is further revealed by a small Tafel slope of 73 mV dec$^{-1}$ (Fig. 3c) and low charge-transfer resistance ($R_{ct}$ = 0.25 Ω) (Supplementary Fig. 8c). To identify the active phase for HzOR, the performance of controlled catalysts including MXene-free NiCo@C/CF, NiCo-free MXene/CF, bare CF and 20% Pt/C casting on CF (Pt/CF) is evaluated under identical conditions (Fig. 3d). The MXene/CF and bare CF are inactive for HzOR. But the presence of MXene contributes greatly to interconnecting NiCo@C arrays on CF and reducing interfacial resistance. It leads to superior electrochemical active surface area (ECSA, 54.25 m$^2$ g$_{cat}$$^{-1}$) and interfacial

conductivity to MXene-free NiCo@C/CF (35.0 m$^2$ g$_{cat}$$^{-1}$, $R_{ct}$ = 1.8 Ω) (Supplementary Figs. 8c, 10). Poisoning the electrode by trace amounts (e.g., 10 mM) of SCN$^-$ induces an immediate and dramatic activity decay, implying the primary role of NiCo as the active phase in catalyzing HzOR (Supplementary Fig. 8d)[38]. When normalized to active mass, the activity of NiCo@C/MXene/CF still exceeds Pt/CF by over 10–35 folds and NiCo@C/CF for 3.6–4.5 times (Supplementary Fig. 8e, f). The NiCo@C/MXene/CF also excels NiCo@C/CF in terms of ECSA-normalized HzOR activity (Supplementary Fig. 11). During the accelerated durability test, this electrode can work steadily for 2000 cycles or 30 h at 100 mA cm$^{-2}$ with a negligible activity loss (Fig. 3e). The electrocatalytic tests at a scan rate of 1.0 mV s$^{-1}$ or against Hg/HgO reference electrode rule out the effect of double-layer charging or reference electrode on catalyst performance (Supplementary Figs. 12a and 13a, b). The ICP analysis reveals a very low residue of Ni, Co, and Ti ions in the electrolyte after long-term HzOR, showing negligible catalyst leaching during electrolysis (Supplementary Table 2). Afterward, post-mortem SEM, TEM, and XPS analysis show that the NiCo@C/MXene/CF well retains the original 3D architecture, nanoarray-based surface and chemical composition, revealing high robustness against long-term HzOR (Supplementary Fig. 14).

**Half-cell HER performance of NiCo/MXene-based electrode.** The NiCo@C/MXene/CF also works efficiently for catalyzing HER. It requires a low overpotential ($\eta$) of 49 and 235 mV to reach the current densities of 10 ($\eta_{j=10}$) and 500 mA cm$^{-2}$ ($\eta_{j=500}$) in 1.0 M KOH, respectively (Fig. 3f and Supplementary Fig. 9b). The alkaline HER on this electrode undergoes a fast Volmer-Heyrovsky pathway with a small Tafel slope of 54.2 mV dec$^{-1}$ (Supplementary Fig. 15a). Poisoning tests by 10 mM SCN$^-$ suggest that the NiCo dominates the HER activity (Supplementary Fig. 15b)[38]. The presence of MXene on NiCo@C/MXene/CF also induces superior ECSA-normalized activity and charge-transfer kinetics to MXene-free

NiCo@C/CF for HER (Supplementary Fig. 15c, d). The HER on NiCo@C/MXene/CF exhibits a high turnover frequency (TOF) of 2.1 s$^{-1}$ at $\eta = 200$ mV and exchange current density ($j_0$) of 1.34 mA cm$^{-2}$, exceeding the NiCo@C/CF by over 3–5.8 times (Supplementary Fig. 16). These results highlight the significance of MXene in improving the per-site electrocatalytic activity of NiCo@C/ MXene/CF. During long-term HER, this electrode can operate steadily for 2000 sweeps or 60 h with 95% current retention at $\eta = 100$ mV in 1.0 M KOH (Supplementary Fig. 17). Afterward, the NiCo@C/MXene/CF also retains the original texture without collapse or leaching (Supplementary Fig. 18 and Table 2). Likewise HzOR, the effect of double-layer charging or reference electrode on catalyst performance is ruled out by conducting the electrocatalytic tests at a scan rate of 1.0 mV s$^{-1}$ or against Hg/HgO reference electrode (Supplementary Figs. 12b and 13c, d). The HER performance of this electrode remains on the top level of reported 3D electrodes under similar alkaline conditions (Supplementary Table 3). Encouragingly, its high activity can be largely maintained in alkaline seawater (pH 13.8), neutral seawater or the electrolyte with seawater pH (8.3) with poor conductivity and ionic strength at high current densities of 400–500 mA cm$^{-2}$ (Fig. 3f, Supplementary Figs. 19, 20a). The NiCo@C/MXene/CF also shows comparable performance with commercial 20% Pt/C in 1.0 M KOH, alkaline or neutral seawater. It could even outperform the Pt/C at the current densities above 120 mA cm$^{-2}$ (Supplementary Fig. 19a, b), making it attractive for large-current hydrogen production. During long-term HER, the NiCo@C/MXene/CF can work for 120 h in both alkaline and neutral seawater, showing good robustness in corrosive seawater (Supplementary Fig. 19c).

**Performance of hybrid seawater electrolyzer.** Hybrid seawater splitting coupling HzOR and seawater-efficient HER shows a dramatic advantage over OWS in reducing the electricity consumption of seawater electrolysis. It requires an ultralow cell voltage of 0.31 V to achieving a high current density of 500 mA cm$^{-2}$ on NiCo@C/MXene/CF under alkaline conditions (pH 13.8) (Fig. 4a). Using neutral seawater for HER raises the cell voltage to 0.42 V at a current density of 400 mA cm$^{-2}$. This activity still far excels the OWS, which has to conquer huge voltages of 1.80–1.82 V for yielding hydrogen at high current densities of 400–500 mA cm$^{-2}$ on the same electrode in 1.0 M KOH. On this basis, an asymmetric HSE is assembled by using the seawater as the catholyte and 1.0 M KOH with 0.5 M hydrazine as the anolyte feed. The NiCo@C/MXene/CF is used as the identical anode and cathode separated by an anion exchange membrane (AEM). It allows for energy-saving hydrogen production from seawater and simultaneous hydrazine degradation in a single cell while harvesting water with a lower salinity from seawater. The asymmetric design ensures hydrogen yielding at high purity while preventing toxic hydrazine to pollute the seawater. It also avoids the interference of HzOR on the performance of HER with overlapped reduction potential, which reduces the HER activity by 1.5–3 folds in alkaline seawater (Supplementary Fig. 21).

For direct electrolysis of neutral seawater, the HSE can reach a high current density of 500 mA cm$^{-2}$ at a low voltage of 1.05 V, reducing by 45.6% as compared to ASE (1.93 V) (Fig. 4b and Supplementary Fig. 9c). It can steadily work for hydrogen production at a rate of 5.6 mol h$^{-1}$ g$_{cat}^{-1}$ below 1.0 V (without $iR$ correction) for over 85 h at a current density of 300 mA cm$^{-2}$ (Fig. 4c). Accordingly, the basic electricity expense is reduced to as low as 2.39 kWh for yielding 1 m$^3$ H$_2$, which is even superior to the theoretical energy demand of OWS (2.94 kWh/m$^3$ H$_2$). The HSE also exhibits rather comparable in seawater pH (8.3) mimicked electrolyte (Supplementary Fig. 20b). Using alkaline seawater with 1.0 M KOH as the catholyte feed in HSE can

further reduce the cell voltage to an ultralow value of 0.7 V at a high current density of 500 mA cm$^{-2}$, cutting by 63.7% relative to ASE (Fig. 4b). No ClO$^-$ generation is detected under such a low potential, allowing the anode corrosion to be fully eliminated regardless of Cl$^-$ crossover (Fig. 4d and Supplementary Fig. 22, 23). As a sharp contrast, the anode is rapidly dissolved by high-concentration ClO$^-$ corrosion, inducing fast failure of ASE in only 6–7 h. Stable electrolysis can proceed below 0.36 V (without $iR$ correction) for over 120 h at 100 mA cm$^{-2}$ in HSE (Fig. 4c). Even at a high current density of 500 mA cm$^{-2}$, the HSE can still steadily work below 1.15 V (without $iR$ correction) for 140 h to yield hydrogen at a fast rate of 9.2 mol h$^{-1}$ g$_{cat}^{-1}$. The electricity expense is reduced to 2.75 kWh/m$^3$ H$_2$, cutting by ~54% relative to ASE working at 2.53 V. The HSE also exhibit superior energy equivalent inputs but lower CO$_2$ equivalent emission to conventional technologies such as alkaline water electrolysis, natural gas steam reforming and recently reported electrochemical methane splitting for hydrogen production (Fig. 4e and Supplementary Table 4), showing a significant advantage in energy efficiency and processing sustainbility[39]. Gas chromatography (GC) validates the yield of high-purity H$_2$ and N$_2$ with a ratio of $ca$. 2: 1 without Cl$_2$ emission (Supplementary Fig. 24). The Faradaic efficiencies (FE) of HER and HzOR are determined to $ca$. 96% and 99%, respectively (Supplementary Fig. 25). This HSE also far excels the state-of-the-art seawater electrolyzer in terms of energy efficiency for hydrogen production (Fig. 4f and Supplementary Table 5).

The performance of HSE is also evaluated by using seawater with various OH$^-$ concentrations (0.0–3.0 M) as the catholyte and 1.0 M KOH containing 0.5 M N$_2$H$_4$ as the anolyte, which may give some clues on the effect of pH gradient over AEM on cell performance. The pH gradient across AEM would be first reduced until OH$^-$ concentrations in the catholyte rising to the same with the anolyte (1.0 M). Meanwhile, the HER activity is improved with the catholyte pH increasing, leading to a fast increase in cell activity (Supplementary Fig. 26). In this case, the direction of pH gradient over AEM is opposite to that of OH$^-$ diffusion, thereby playing no significant role in electrocatalytic enhancement. After the catholyte pH exceeds the anolyte value, the direction of pH gradient across AEM would be the same with OH$^-$ diffusion. But the performance of the electrolyzer is not significantly enhanced at large current densities due to the limitation of AEM in ionic exchange capacity and permeability. Developing high-performance AEM is desired to address this issue for full exploitation of the potential of our hybrid electrolyzer design.

Hydrazine is strongly poisonous and carcinogenic with a high risk to human health. The threshold of hydrazine in surface water should be strictly restricted to protect the ecosystem. The HSE coupling HzOR provides the additional function in dealing with toxic hydrazine sewage without using extra oxidants or complex separation technologies. It allows the hydrazine to be removed at a fast rate of $4.34 \pm 0.007$ mol h$^{-1}$ g$_{cat}^{-1}$ during hydrogen production at a high current density of 500 mA cm$^{-2}$ (Fig. 4g and Supplementary Fig. 27). The residual limit of hydrazine in water can be as low as 3 ppb, falling below the allowable value (10 ppb) set by EPA[30]. Exceptionally stable hydrazine treatability and high hydrogen-yield rate can be maintained for repeated cycles, showing high effectiveness for practical use. Using hydrazine sewage as the anolyte may in turn further reduce the hydrogen cost while applying costless seawater as the catholyte feed. On this basis, cost-effective and sustainable hydrogen production might be scaled up by feeding costless seawater and industrial hydrazine sewage into renewables-powered HSE in the coastal region with intense solar irradiation and strong wind pattern (Fig. 4h).

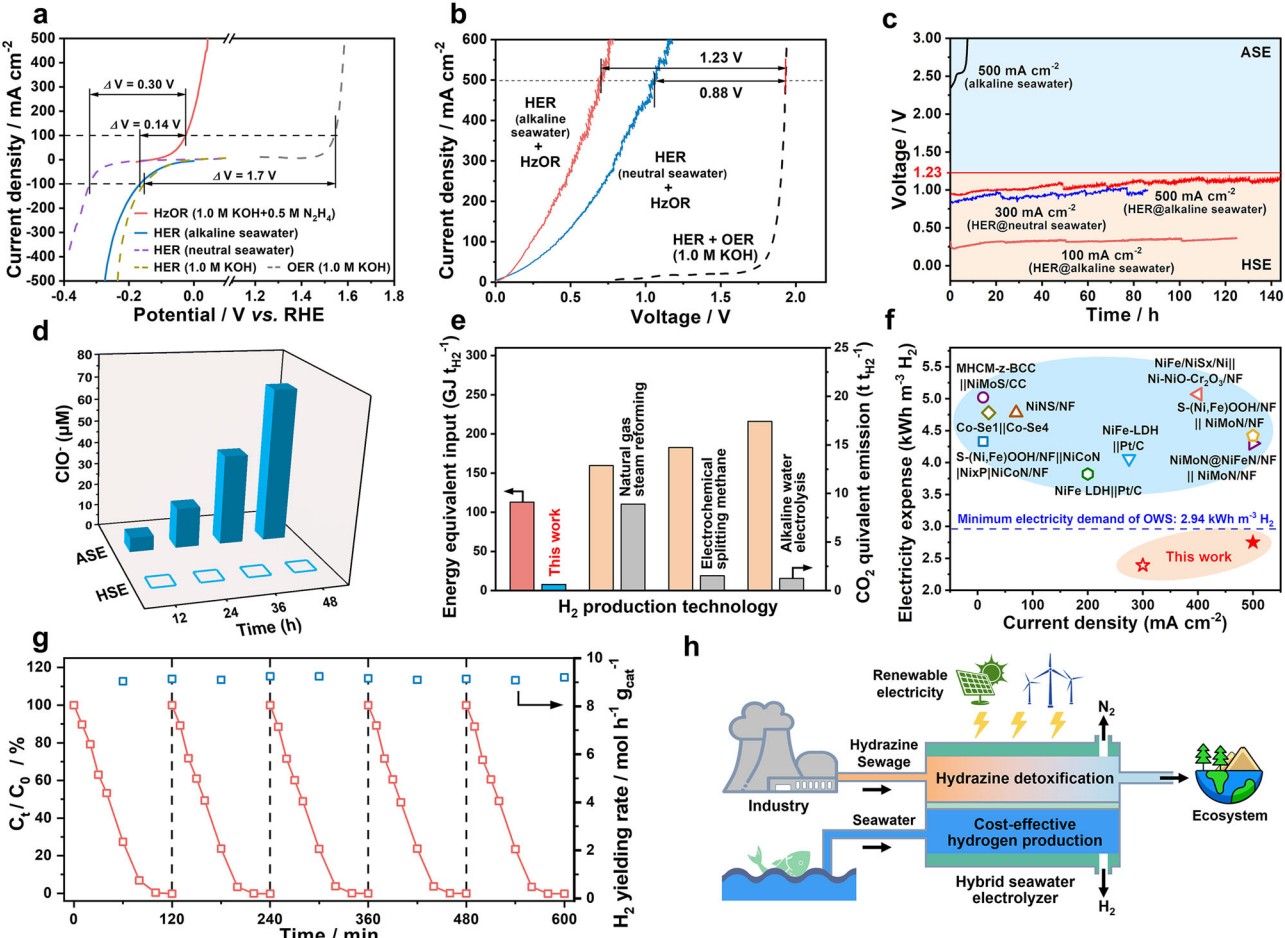

**Fig. 4 Performance of HSE for hydrogen production and hydrazine degradation. a** The voltage differences (Δ V) between HER and HzOR or OER on NiCo@C/MXene/CF in different electrolytes. **b** The LSV curves of HSE using neutral or alkaline seawater as the catholyte, compared with ASE. **c** Durability tests of HSE at various current and catholyte conditions. The ASE is also tested at 500 mA cm$^{-2}$ for comparison. **d** A comparison of the ClO$^-$ concentration change in the anolyte during continuous electrolysis at 100 mA cm$^{-2}$ in HSE or ASE. **e** A comparison of HSE with different hydrogen production techniques in energy equivalent input and CO$_2$ equivalent emission. **f** A comparison of HSE (champagne region) with the state-of-the-art seawater electrolyzer (pale blue region) in cell voltage and current density. **g** The activity and durability of HSE for hydrazine degradation during hydrogen production at 500 mA cm$^{-2}$. **h** Schematic drawing of cost-effective and sustainable hydrogen production by renewables-powered HSE with costless seawater and industrial hydrazine sewage as the feeds.

**Self-powered system for low-voltage hydrogen production from seawater.** Large-scale renewable energy systems are usually necessary to satisfy the high power demand of water electrolysis. While the HSE with low electricity expense offers the feasibility of integrating with renewable power sources on a smaller scale (e.g., fuel cells, solar cells). This benefit is highly desired to reduce the additional capital and complexity of the hydrogen production system. As a proof-of-concept, a self-powered hydrogen production system is built by integrating the HSE into a single direct hydrazine fuel cell (DHzFC, Fig. 5a, b). The DHzFC is assembled by using NiCo@C/MXene/CF as the anode and 20% Pt/C as the cathode. It exhibits an open-circuit voltage (OCV) of *ca.* 1.0 V and a maximum power density of 53.5 mW cm$^{-2}$ (Supplementary Fig. 28). This self-powered system could yield hydrogen from seawater at a rate of 1.6 mol h$^{-1}$ g$_{cat}$$^{-1}$ with hydrazine as the sole energy consumable (Fig. 5c and Supplementary Movie 1). It achieves a total efficiency of 48.0%, comparable with the reported systems for converting hydrazine to hydrogen[25,26]. Hydrogen production with better sustainability and cost-effectiveness can be realized by connecting the HSE into photovoltaic cells powered by easily harvestable and clean solar energy (Fig. 5a, d). Such a solar-driven hydrogen production system could be operated at a

current density of *ca.* 310 mA cm$^{-2}$ and an average photovoltage of *ca.* 0.876 V when powered by a single commercial solar cell (1.0 W) (Supplementary Fig. 29). The hydrogen is yielded at a decent rate of 6.0 mol h$^{-1}$ g$_{cat}$$^{-1}$ from seawater under AM 1.5 G simulated solar illumination with a power density of 100 mW cm$^{-2}$ (Fig. 5e, h and Supplementary Movie 2). Under natural light, this system can also work steadily for efficient hydrogen production (Supplementary Movie 3).

## Discussion

**Origin of the intrinsic activity of NiCo sites for promoting HzOR.** Regarding the extensive investigation in HER on NiCo alloy, we mainly focus on the origin of its HzOR activity. Typical (100), (110), and (111) facets of fcc Ni$_3$Co alloy are studied for the first-principle calculation. All these facets exhibit strong chemical interaction with N$_2$H$_4$ molecule via the N-metal donor-acceptor pairs with rather negative binding energy ($E_b$) of −1.24 to −1.89 eV (Fig. 6a and Supplementary Table 6). The N$_2$H$_4$ molecule is attracted on Ni$_3$Co surface in three configurations: on the top sites of Ni or Co atoms, or the bidentate sites between them. The (100) facet of Ni$_3$Co alloy shows the strongest N$_2$H$_4$ adsorption with the most negative $E_b$ of −1.54 to −1.89 eV (Supplementary Table 6).

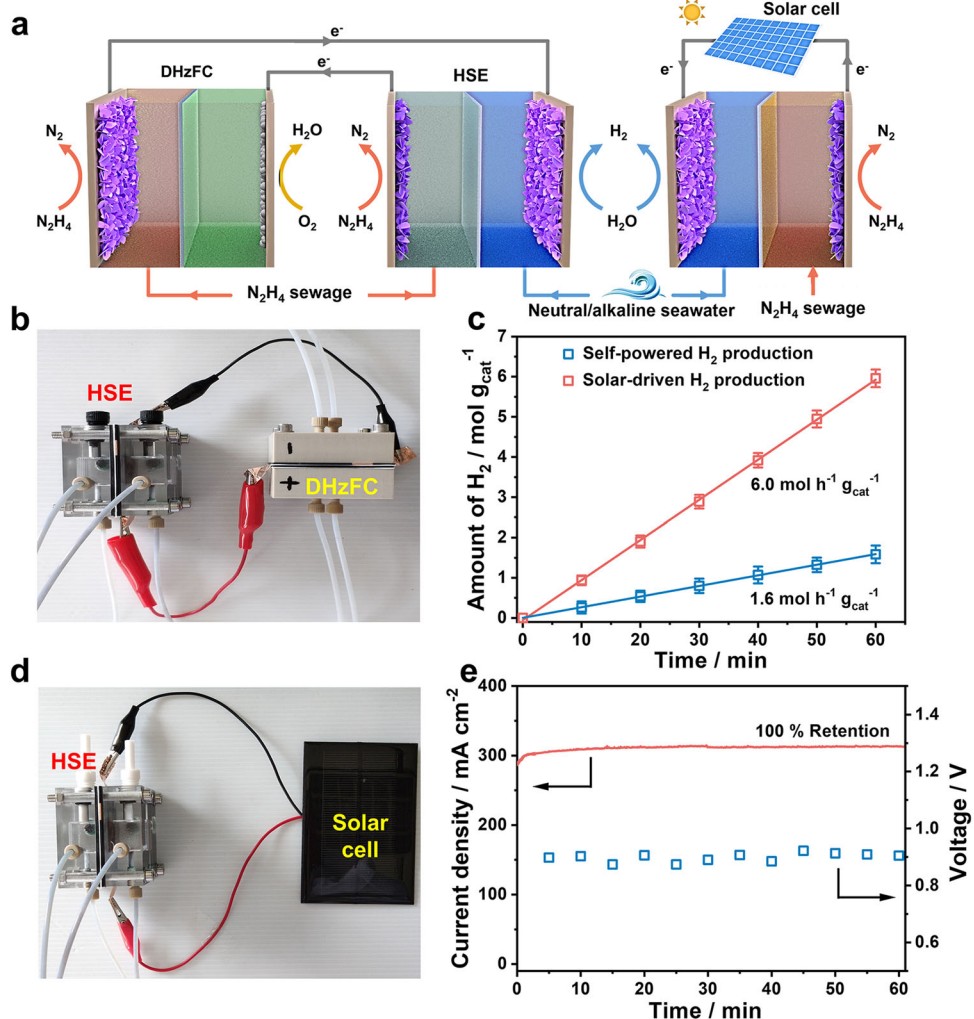

**Fig. 5 Self-powered hybrid seawater electrolysis systems. a** Schematic illustration of self-powered hydrogen production systems by integrating HSE to low-voltage DHzFC or solar cell. **b** Optical image of a self-powered hydrogen production system connecting an HSE to a DHzFC. **c** Hydrogen-yield rate of an HSE powered by a single DHzFC or solar cell. **d** Optical image of a solar-driven HSE. **e** Current density or voltage vs. time curves of this solar-driven hydrogen production system under AM 1.5 G illumination.

Overall, the strongest $N_2H_4$ adsorption is achieved by the bidentate-type attraction between both N atoms in $N_2H_4$ molecule and Ni and Co sites. Charge density difference analysis indicates prominent charge transfer from the N atoms in $N_2H_4$ to nearby Co and Ni atoms (Fig. 6a), which enlarges the N-H bond length in $N_2H_4$ adsorbed on $Ni_3Co$ alloy to 1.028–1.035 Å relative to the free molecule (1.024–1.026 Å). The effect weakens the N-H bonds in adsorbed $N_2H_4$ molecule to facilitate the molecular activation for HzOR[26,40]. Among all facets of $Ni_3Co$ alloy, the (100) plane exhibits the highest activity for activating $N_2H_4$ molecule, as identified by the longest length of N-H bonds (1.033–1.035 Å) (Fig. 6a). The HzOR in alkaline medium may undergo three possible pathways, namely, the 1e route with $NH_3$ and $N_2$ as the product ($N_2H_4 + OH^- \rightarrow NH_3 + 0.5N_2 + H_2O + e^-$), the 2e pathway yielding $N_2$ and $H_2$ ($N_2H_4 + 2OH^- \rightarrow N_2 + H_2 + 2 H_2O + 2e^-$), and 4e reaction releasing $N_2$ ($N_2H_4 + 4OH^- \rightarrow N_2 + 4H_2O + 4e^-$). Since high-purity $N_2$ is detected as the only anodic product by GC (Supplementary Fig. 24b), a 4e pathway should be responsible for HzOR on our catalyst[28,40]. On this basis, the elementary reactions for stepwise $N_2H_4$ dehydrogenation ($N_2H_4 \rightarrow N_2H_4^* \rightarrow N_2H_3^* \rightarrow N_2H_2^* \rightarrow N_2H^* \rightarrow N_2^*$) are investigated on three facets of $Ni_3Co$ alloy (Fig. 6b and Supplementary Fig. 30). The $N_2H_4$ adsorption on all these facets is thermodynamically spontaneous and the initial

dehydrogenation to $N_2H_3^*$ and $N_2H_2^*$ is endothermic. The initial dehydrogenation of $N_2H_4^*$ to $N_2H_3^*$ encounters a high energy barrier of 0.7–0.8 eV on the (111) and (110) facets in contrast to a much lower value of 0.28 eV on the (100) facet. Next dehydrogenation step of $N_2H_3^*$ to $N_2H_2^*$ is also uphill on all facets with a comparable energy barrier of 0.3–0.5 eV, followed by the downhill steps of $N_2H_2^*$ dissociation to $N_2H^*$ and $N_2^*$. These findings suggest the dehydrogenation of $N_2H_3^*$ to $N_2H_2^*$ is the potential rate-limiting step of HzOR on $Ni_3Co$ (100) facet while that for (110) and (111) facets is the $N_2H_4^*$ to $N_2H_3^*$ with a much higher energy barrier. Therefore, the $Ni_3Co$ (100) facet is predicted to be the most active for propelling HzOR.

**Effect of interfacial properties on electrolysis performance.** Besides the NiCo alloy, the MXene also contributes greatly to tailoring the interfacial properties for boosting the performance of NiCo@C/MXene/CF. The $Ti_3C_2T_x$ MXene has an even superior conductivity (*ca.* 5600 S cm$^{-1}$) to soft carbon materials (e.g., 10–120 S cm$^{-1}$ for reduced graphene oxide, 100 S cm$^{-1}$ for carbon black)[41]. Their presence significantly reduces the interfacial resistance to accelerate the charge-transfer kinetics of both HER and HzOR on NiCo@C/MXene/CF (Supplementary Fig. 8c, 15c). Besides, the presence of abundant -OH and -O groups on MXene

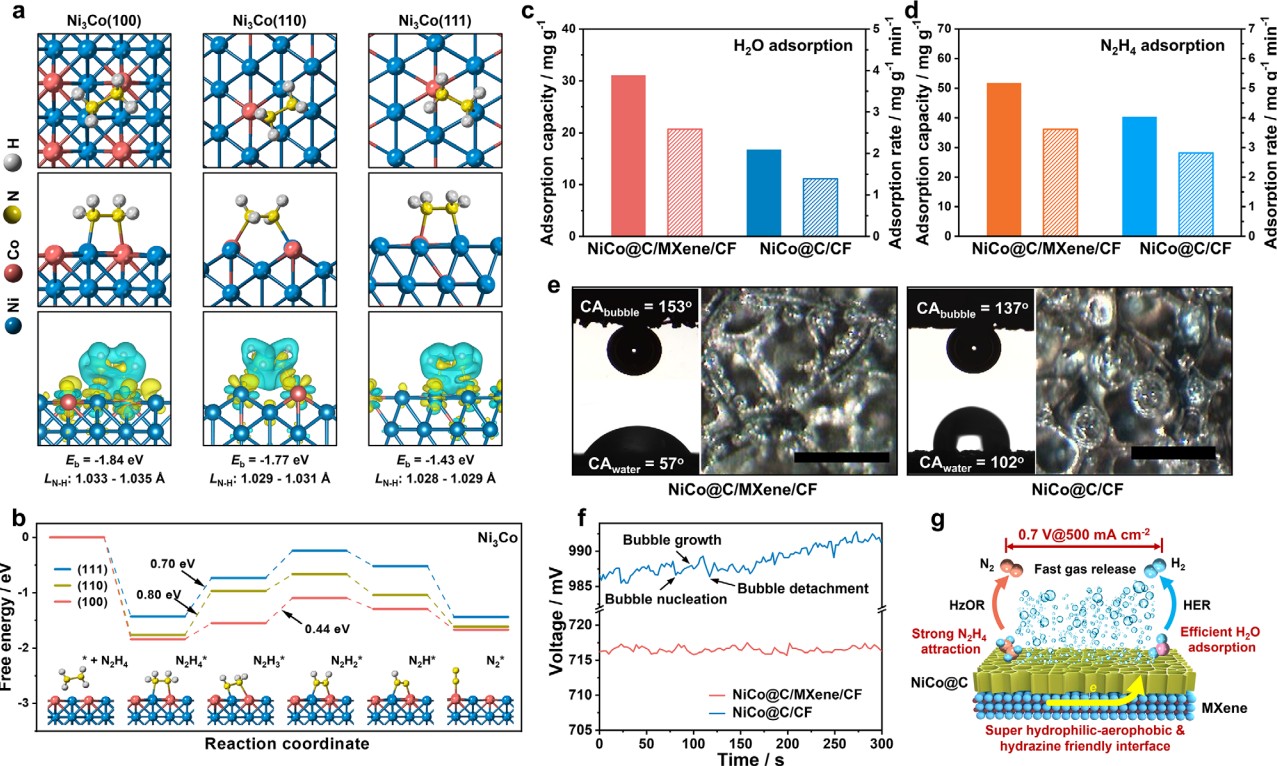

**Fig. 6 Role of NiCo alloy and interfacial properties in promoting electrocatalytic performance. a** The structural model of $N_2H_4$ adsorption on various facets of $Ni_3Co$ alloy, and corresponding charge density difference analysis, where the yellow or cyan regions indicate the accumulation or depletion of the charge, respectively. The $L_{N-H}$ and $E_b$ are the calculated N-H bond lengths (Å) and binding energy of the intermediates on $Ni_3Co$ alloy. **b** Free energy profiles of stepwise HzOR on different facets of $Ni_3Co$ alloy. Inset is the corresponding structural evolution of reaction intermediates adsorbed on the (100) facet of $Ni_3Co$. Adsorption capability of (**c**) water or (**d**) hydrazine on NiCo@C/MXene or NiCo@C. **e** The contact angels of water ($CA_{water}$) and bubble ($CA_{bubble}$) on NiCo@C/MXene/CF or NiCo@C/CF, and the optical images of gas bubbles released from both electrodes during HER. Scale bars: 500 μm. **f** Chronopotentiometric curves of NiCo@C/MXene/CF and NiCo@C/CF for hybrid seawater electrolysis. **g** Schematic illustration of the electrocatalytic enhancement of NiCo@C/MXene/CF at large current densities by overall enhancement in interfacial properties in terms of conductivity, robustness, water/hydrazine adsorption, and gas-releasing capability.

surface also facilitates the effective attraction of water and hydrazine molecules onto the electrocatalytic interface by hydrogen bonding attraction[42]. Efficient water adsorption on NiCo@C/MXene/CF is revealed by a small water contact angle ($CA_{water} = 57°$), superior water adsorption capacity (31.0 mg $g_{cat}^{-1}$) and rate (2.6 mg $min^{-1}$ $g_{cat}^{-1}$) to hydrophobic NiCo@C/CF (Fig. 6c and Supplementary Fig. 31). This improvement is important to accelerate the Volmer-Heyrovsky kinetics of neutral/alkaline HER (Supplementary Fig. 15a), where the water supplies the hydrogen and its dissociation (Volmer step) is the rate-limiting step[3,43]. The NiCo@C/MXene/CF also shows superior hydrazine adsorption capacity (51.6 mg $g_{cat}^{-1}$) and rate (3.6 mg $min^{-1}$ $g_{cat}^{-1}$) to NiCo@C/CF (40.3 mg $g_{cat}^{-1}$, 2.8 mg $min^{-1}$ $g_{cat}^{-1}$), which contribute greatly to a 2-folds faster HzOR kinetics (Figs. 3c, 6d). Such highly hydrophilic and hydrazine-friendly properties are essential to achieving high access of water or hydrazine molecules to the inner Helmholtz plane above the electrocatalytic interface[44]. It is undoubtedly important to accelerate the HER and HzOR, where water or hydrazine plays a vital role in governing the overall kinetic performance.

A common feature of HzOR and HER is the intense gas-releasing characteristic. High coverage of gas bubbles on the electrode surface severely blocks the ECSA and induces high ionic diffusion resistance, causing considerably higher ohmic drop than contributed by other factors at high current densities[15,45]. The overpotential caused by this problem is a dominant factor limiting the water electrolysis performance at high current

densities[46]. Besides, the bubble growth, collapse and detachment from the electrode generate large mechanical strain and stretch force to destruct the catalyst and deteriorate the cell performance[47]. The bubble problem is even worse for hybrid seawater splitting with intense gas releasing from both anode and cathode. This difficulty is mitigated by engineering a NiCo@C/ MXene/CF electrode with macroporous gas transport channels and a nanoarray-based superaerophobic surface. The electrode with a huge bubble contact angle ($CA_{bubble} = 153$ °) could effectively facilitate the rapid release of small gas bubbles (<60–80 μm) during electrolysis (Fig. 6e and Supplementary Fig. 32). This improvement is critical to maintaining a sufficient electrode-electrolyte-gas triple-phase interface for stable operation of electrolysis under vigorous gas-releasing conditions (Fig. 6f)[33,45,48]. Without MXene, irregular microstructures are formed on the surface of NiCo@C/CF due to poor chemical coupling between NiCo@C and CF (Supplementary Fig. 1i). Such structural degradation largely reduces not only the ECSA but also the aerophobic properties of the electrode (Fig. 6e and Supplementary Fig. 10). Accordingly, the NiCo@C/CF encounters huge voltage fluctuation caused by vigorous accumulation and detachment of large gas bubbles, which reduces hydrogen yielding efficiency. These results suggest that an overall enhancement in interfacial conductivity, robustness, water/hydrazine adsorption and gas-releasing capability is vital to boosting the electrocatalytic enhancement of NiCo@C/MXene/CF for hybrid seawater splitting at large current densities (Fig. 6g).

In conclusion, an efficient strategy coupling seawater reduction with thermodynamically favorable hydrazine oxidation is developed to address two extreme challenges of seawater electrolysis: the huge electricity consumption and notorious anode corrosion by chlorine chemistry. A NiCo/MXene-based electrode with superaerophobic-hydrophilic and hydrazine-friendly electrocatalytic interface is designed to fully exploit the potential of this chemistry. It allows for overall enhancement in interfacial conductivity, robustness, water/hydrazine adsorption capability and gas-releasing pattern for boosting electrolysis performance at large current densities. The hybrid seawater electrolyzer enables hydrogen production at ultralow cell voltages of 0.7–1.0 V, which fully avoids the chlorine hazards on cell performance in neutral or alkaline seawater. Meanwhile, the hydrogen can be produced at an intense rate of 9.2 mol h$^{-1}$ g$_{cat}^{-1}$ by stable seawater electrolysis for 140 h at 500 mA cm$^{-2}$ with high Faradaic efficiency. The electricity expense is largely reduced by 30–52% at a high current density of 500 mA cm$^{-2}$ relative to commercial alkaline water electrolysis and the state-of-the-art seawater electrolyzers. Simultaneously, rapid hydrazine degradation to a rather low residual of ~3 ppb can be achieved at a fast rate of 4.34 ± 0.007 mol h$^{-1}$ g$_{cat}^{-1}$. Self-powered hybrid seawater electrolysis is also realized by integrating hydrazine fuel cells or solar cells for hydrogen production with better sustainability. Our work may show the practical impact on the efficient utilization of hydrogen reserved with unlimited abundance in the ocean for approaching carbon-neutral hydrogen economy.

## Methods

**Fabrication of NiCo@C/MXene/CF**. The MXene/CF was first prepared by immersing copper foam pretreated by 1.0 M HCl into Ti$_3$C$_2$T$_x$ MXene colloids (5 mg mL$^{-1}$), followed by drying in vacuum at 40 °C for 12 h. A piece of MXene/CF (2 × 3 cm$^2$) and 2, 6-naphthalenedicarboxylic acid dipotassium (0.2 mmol) were added into a aqueous solution (25 mL) of Ni (CH$_3$COO)$_2$·4H$_2$O (0.1 mmol) and Co (CH$_3$COO)$_2$·4H$_2$O (0.1 mmol). The reaction was conducted in a sealed autoclave at 80 °C for 3 h. The obtained product was annealed at 400 °C for 2 h at a ramp rate of 5 °C min$^{-1}$ in NH$_3$ flow to yield NiCo@C/MXene/CF. As a control sample, the NiCo@C/CF was also prepared in a similar way in the absence of MXene. The Pt/CF electrode with an average mass loading of *ca.* 1.0 mg cm$^{-2}$ was also made by casting the catalyst ink (125 μL) of commercial 20% Pt/C (1.0 mg) in ethanol, DI water and Nafion (5.0 wt.%) with a volume ratio of 100: 97: 3 on the CF (1 × 1 cm$^2$).

**Material characterization**. The SEM and TEM images were taken with a field-emission scanning electron microscopy (FEI NanoSEM 450) and transmission electron microscopy (FEI TF30). The X-ray diffraction (XRD) analysis was done on a Bruker D8 Advance X-ray spectrometer equipped with a 2D detector (Cu Kα, λ = 1.5406 Å). The Raman analysis was conducted on a Thermo Fisher Scientific DXR Raman microscopy using laser excitation (λ = 532 nm). The UV-vis analysis was performed on a UV-vis-NIR spectrometer (PerkinElmer Lambda 750). The XPS measurements were performed using Thermo ESCALAB MK II X-ray photoelectron spectrometer with C 1s (284.6 eV) calibration. The weight ratio of the elements in the sample was determined by inductively coupled plasma optical emission spectroscopy (ICP-OES, Optima 2000DV, PerkinElmer). The water contact angle was measured by using a contact angle goniometer (SL150E, USA KINO). The contact angle of the gas bubble in the electrolyte was measured using the OCA21 Dataphysics instrument via the captive-bubble method controlled bubble volume to 2 μL. The optical images of gas bubbles were recorded by a high-speed CCD camera (i-SPEED 3, AOS Technologies) equipped with an optical microscope (SZ-CTC, OLYMPUS). The water/hydrazine adsorption capability was measured by a QCM 200 electrochemical quartz crystal microbalance (EQCM). The XAFS analysis was conducted in Shanghai Synchrotron Radiation Facility (SSRF).

**Half-cell HzOR test**. The tests were conducted on a CHI 760E electrochemical workstation with a standard three-electrode system. The as-prepared samples were directly used as the working electrode with an average NiCo@C loading of *ca.* 1.0 mg cm$^{-2}$, while a graphite rod and an Ag/AgCl electrode (saturated with KCl solution) were employed as the counter and reference electrode, respectively. The electrolyte was 1.0 M KOH containing 0.5 M hydrazine. All the tests were maintained in Ar-saturated electrolyte throughout the test period. The linear sweep voltammetry (LSV) curves were recorded from −1.2 to −0.5 V (vs. Ag/AgCl) at a scan rate of 10 mV s$^{-1}$. Accelerated durability tests were conducted through continuous potential cycling ranged from −1.2 to −0.5 V (vs. Ag/AgCl) at a scan rate

of 50 mV s$^{-1}$. Chronopotentiometric tests were recorded by applying a current density of 100 mA cm$^{-2}$. The AC impedance measurements were carried out at a potential of −0.9 V (vs. Ag/AgCl) in a frequency range from 100 kHz to 1 Hz by applying an AC voltage with 5 mV amplitude. All potentials measured were converted to the value against RHE according to the equation:

$$E_{RHE} = E_{Ag/AgCl} + 0.059 \times pH + 0.197 \tag{1}$$

All the measurements were performed with *iR* compensation except for the chronopotentiometric tests.

**Half-cell HER test**. The HER tests were conducted by using a similar electrode configuration and workstation with HzOR measurement. The electrolyte was 1.0 M KOH or seawater containing 1.0 M KOH or neutral seawater. For HER in seawater, natural seawater with pH 8.3 was collected from the Bohai Sea (Dalian, China), which was filtered to remove visible impurities before use. All the tests were maintained in Ar-saturated electrolyte throughout the test period. The LSV curves were performed from −0.9 to −1.6 V (vs. Ag/AgCl) at a scan rate of 10 mV s$^{-1}$. The AC impedance measurements were carried out at an overpotential of 200 mV over the frequency range of 100 kHz to 1 Hz and an amplitude of 5 mV. Accelerated durability tests were performed in a potential range of −0.9 to −1.5 V (vs. Ag/AgCl) at a scan rate of 50 mV s$^{-1}$. Chronoamperometric measurement was conducted at a controlled overpotential. All potentials measured were converted to the value against RHE according to Eq. (1). All the measurements were performed with *iR* compensation except for the chronoamperometric tests.

**Assembly and tests of HSE**. The hybrid seawater electrolyzer was assembled by using a homemade two-electrode flow cell with NiCo@C/MXene/CF with the same area as the identical anode and cathode. The seawater containing 1.0 M KOH or neutral seawater was used as the catholyte while 1.0 M KOH with 0.5 M hydrazine was fed as the anolyte, which were cycled by the peristaltic pumps (Longer, BT100-2J). The cathode and anode chambers were separated by an anion exchange membrane (Fumasep FAA-3-PK-130). The electrolysis tests were performed on a CHI 760E electrochemical workstation. The polarization curves were measured at a scan rate of 10 mV s$^{-1}$ with *iR* compensation. The stability test was carried out at the controlled current densities. The gas products from the cells were collected and examined by gas chromatography (GC, Agilent Technologies 7890 N). Meanwhile, the theoretical amount of evolved gas can be calculated by the equation:

$$N = I \times t/(n \times F) \tag{2}$$

where the N is the theoretical amount (mol) of evolved gas after electrolysis for a certain time (t) at a fixed current (I), n is the number of electrons transferred (n = 2 for HER, n = 4 for HzOR), F is the Faraday constant (96485 C mol$^{-1}$). The Faradaic efficiency (FE) can be estimated according to the ratio of the measured to the theoretical gas amount.

**Measurement of hydrazine degradation rate**. The hydrazine content in the electrolyte was determined by the Watt and Chrisp method[49]. A mixture of para-(dimethylamino) benzaldehyde (5.99 g), concentrated HCl (30 mL) and ethanol (300 mL) was used as the color reagent. During electrolysis, a part of the anolyte was periodically collected and quantitatively diluted with a stock solution of 1.0 M HCl (10 mL), followed by adding the color reagent (5 mL) under stirring for 20 min. The hydrazine concentration was determined by the UV-vis spectrum of the obtained solution at λ = 457 nm. The concentration-absorbance curve was calibrated using a standard solution containing hydrazine monohydrate with a series of concentrations in 1.0 M KOH by a correlation of y = 1.3037 x + 0.0014 (R$^2$ = 0.9999). The removal rate (k) of hydrazine was calculated by the equation:

$$k = (a_0 - a_t)/(m \times t) \tag{3}$$

where $a_0$ and $a_t$ are the initial and final amount of hydrazine after electrolysis for a certain time (t), and m is the mass loading of NiCo@C (g).

**Assembly of the self-powered hydrogen production systems**. The direct hydrazine fuel cell was built by using NiCo@C/MXene/CF as the anode and 20% Pt/C loaded on carbon paper as the cathode. Two electrodes were separated with a Nafion 117 membrane. The catholyte was O$_2$-saturated 0.5 M H$_2$SO$_4$ and the anolyte was 1.0 M KOH containing 0.5 M N$_2$H$_4$ that were fed into the cell at a flow rate of 10 mL min$^{-1}$. This fuel cell was connected to the above HSE for constructing a self-powered hydrogen production system with hydrazine as the sole fuel consumable. The total efficiency (TE, %) of this system was estimated by the equation:

$$TE = N_{H2}/(2 \times N_{N2H4}) \tag{4}$$

where $N_{N2H4}$ and $N_{H2}$ are the amount of hydrazine consumed (mol) and H$_2$ produced (mol), respectively. The solar-powered hydrogen production system was built by connecting such electrolyzer to a commercial Si solar cell (8 × 11 cm$^2$, 1 W) powered by simulated sunlight (AAA solar simulator, 94032 A, Newport, US) or natural solar light. The Keithley 2450 source meter and multimeter were used to measure the current and voltage in the circuit, respectively.

## Data availability
Source data are provided with this paper. Extra data that support the findings of this study are available from the corresponding author upon reasonable request. Source data are provided with this paper.

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

## Acknowledgements
We sincerely thank Dr. Wenxin Chen at the Beijing Institute of Technology for his kind help on XAFS analysis. This work was supported by the National Natural Science Foundation of China (NSFC, No. 51772040, 51972040, 51522203), Talent Program of Liaoning (No. XLYC1807032), Innovation Program of Dalian City (No. 2018RJ04) and the Fundamental Research Funds for the Central Universities (No. DUT20TD203, DUT20LAB307).

## Author contributions
Z.Y.W., F.S., and J.S.Q. (Prof. Jieshan Qiu) conceived the idea and co-wrote the paper. F.S. and J.S.Q. (Ms. Jingshan Qin) performed the experiments and theoretical calculations. M.Z.Y. and X.H.W. helped with the material characterization. Z.Y.W., X.M.S., and J.S.Q. (Prof. Jieshan Qiu) guided all aspects of the work.

## Competing interests
The authors declare no competing interests.
