## [Peer Review File · Nature Communications]

REVIEWER COMMENTS

Reviewer #1 (Remarks to the Author):

High energy consumption and chlorine evolution/corrosion are two critical obstacles to the practical deployment of seawater electrolysis technology for a long period. This work by Qiu and co-workers offers a rather good opportunity to breaking both bottlenecks. New chemistry of hybrid seawater splitting enables impressive performance in reducing electricity consumption and material cost for affordable hydrogen production while treating hydrazine pollution effectively at industrially large current. Successful demonstration of self-powered electrolyzers further indicates the potential of this technology for sustainable hydrogen production by renewables-driven seawater electrolysis. Systematic experiments and theoretical calculations have been conducted to characterize the catalyst and catalytic mechanism in a well-organized form. I would like to recommend the acceptance of this work with a minor revision according to the following points:

1. The authors should compare the HER activity of the present catalyst with commercial Pt/C under similar conditions. The latter is a standard catalyst for HER.
2. Whether the reference electrodes were calibrated against RHE? Necessary correction should be conducted if not.
3. The calculation was performed for a pathway of $N_2H_4 \rightarrow N_2H_4^* \rightarrow N_2H_3^* \rightarrow N_2H_2^* \rightarrow N_2H^* \rightarrow N_2^*$. Are there any other possible reaction ways especially those involving NH_3 ? The authors are suggested to provide related discussions with supporting calculation results.
4. In Fig. 4c, what is the meaning of ASE and OSS, as well as HSE and HSS? The authors should clearly indicate these abbreviations.
5. A very recent benchmark work (Angew. Chem. Int. Ed. 2021, 60, 5984) on electrocatalytic hydrazine oxidation can be cited to enrich the background of this work.
6. A typo error, the "Mm" in the y-axis in Fig. 4d, needs to be corrected.
7. In Fig. 4g, the hydrogen yielding rate kept stable but hydrazine concentration was rapidly reduced in the same electrolyzer. Can the author explain it?

Reviewer #2 (Remarks to the Author):

Comments: Minor revision

Overall: This is a very interesting paper showing thermodynamically favorable hydrazine oxidation to address two extreme challenges of seawater electrolysis: the huge electricity consumption and anode corrosion by chlorine chemistry. They have shown high a faradaic current efficiency with less power consumption. The idea of using hydrazine oxidation rather than water is novel and the idea can be used in other areas. The article can be published after addressing the following comments:

In "Introduction"

- 1) "Efficient technologies are thus highly desired to degrade the hydrazine in surface water to a trace residual" (page 4). What is the typical hydrazine concentration in wastewater and are they enough to produce energy for practical applications?

If hydrazine is added then the authors need to do cost analysis to show the actual cost-effectiveness of the hybrid seawater electrolyzer system over conventional ones as they have used hydrazine, a value added product.

- 2) Seawater and wastewater matrices are completely two different systems. This is unfeasible to use them together.

In Results

- 1) "While the MXene layer with abundant -OH, -O and -F groups may effectively attract the water and hydrazine molecules via hydrogen bonding". -F groups don't do hydrogen bonding effectively.

In "Half-cell HzOR performance of NiCo/MXene-based electrode"

- 1) "The HzOR activity of NiCo@C/MXene/CF is evaluated in 1.0 M KOH with various hydrazine

concentrations". Why did the authors use 1.0 M KOH when their electrolyzer is based on seawater with neutral pH? Reaction mechanism depends on solution pH critically. The authors need to do all the experiments (OER+HER) in seawater pH mimicked electrolyte and calculate all the parameters. In "Half-cell HER performance of NiCo/MXene-based electrode"

1) The authors stated the usefulness of the use of Mxene. However they need to state at least some reasons or hypothesis for its better activity. Is it only conductivity?

In "Performance of hybrid seawater electrolyzer"

1) "seawater as the catholyte and 1.0 M KOH with 0.5 M hydrazine as the anolyte feed." As two different solution pH is on the both sides of the electrolyzer how does the pH gradient over the membrane affect the performance?

2) The authors have missed couple of significant work by other authors:

Dresp, S., Thanh, T. N., Klingenhof, M., Brückner, S., Hauke, P., & Strasser, P. (2020). Efficient direct seawater electrolyzers using selective alkaline NiFe-LDH as OER catalyst in asymmetric electrolyte feeds. *Energy & Environmental Science*, 13(6), 1725-1729.

Gayen, P., Saha, S., & Ramani, V. (2020). Selective seawater splitting using pyrochlore electrocatalyst. *ACS Applied Energy Materials*, 3(4), 3978-3983.

Reviewer #3 (Remarks to the Author):

In this manuscript the authors coupled hydrazine oxidation at the anode with sea water splitting at the cathode. This yields hydrogen at a very low cell voltage which is impressive. They have used NiCo alloy on MXene coated Cu foam as a catalyst. The result is good mainly because thermodynamic potential of hydrazine oxidation (HzOR) is very low compared to water oxidation and not due to selectivity of the catalyst. Previously also urea oxidation instead of water oxidation is chosen as anodic reaction to lower the cell voltage. Moreover, the Cl₂ evolution from sea water is not prominent in alkaline medium according to the previous reports. So, here also catalyst is not playing any prominent role from the 'result' point-of-view. I could not find any novelty, conceptual advancements, rational and intellectual thinking in terms of choosing the material. This work is not suitable for publication in Nature communications. Following are the comments to improve quality of the work before submission elsewhere.

1. During synthesis of the final catalyst, why NH₃ is used for annealing? The annealing in presence of NH₃ can form NiCo nitride instead of the alloy. Moreover, XRD reflections of NiCo alloy and nitride appears at similar position. So why it is an alloy and not its nitride?

2. The role of MXene in enhancing the conductivity and ECSA is not clear. It is mentioned that MXene is more conducting than soft carbon. But NiCo alloy is generated from MOF, so it is already assisted with conducting graphitic carbon which is shown by Raman spectra. What is the need of MXene? Why MXene enhances ECSA, whereas it is catalytically inert? What is the activity of MXene on copper foam towards HzOR and HER?

3. In Fig. 2d, is it a single nanosheet or aggregation of nanoparticles? Where are the lattice fringes of MXene? Is it amorphous?

4. Perform AFM imaging of NiCo@C/MXene/CF, NiCo@C/CF and MXene/CF to show the thickness and lateral length of each nanosheet.

5. I will recommend EXAFS and XANES analyses of the final catalyst as well as Ni@C/MXene and Co@C/MXene to show the Ni-Co binding and zero oxidation state of the alloy and change of coordination on going from individual metals to the alloy phase.

6. In Figure S2, the XRD reflections of Cu foam also appear at almost the same position, how the authors could distinguish the peaks from the substrate and the catalyst? Does peeling of the catalyst eliminate all Cu? XRD pattern of bare Cu foam and also NiCo@C/MXene on Cu foam is recommended. Then the XRD of NiCo@C/MXene on Cu foam should also show two consecutive peaks, one due to the catalyst, another due to Cu foam. Where is the XRD pattern of individual metallic Co and Ni in Ni@C/MXene and Co@C/MXene?
7. In Figure S3, why the maximum binding is for Ti^{3+} and not due to Ti-C? Ar sputtering might help in increasing the intensity of Ti-C binding.
8. Is Pt state-of-the-art catalyst for HzOR? Why the activities for this reaction compared with Pt?
9. In Figure 3a, what is the reason for negative current below -0.1V to -0.33V?
10. Conducting LSV at 10 mV/s scan rate for any reaction exaggerates the activity. The authors should perform all the electrocatalytic measurements of the final catalyst at a lower scan rate like 1 mV/s.
11. What is the reason for noisy LSV plot for NiCo@C/CF?
12. In Figure S9c and d, deconvolution of XPS peak of Ni and Co 2p should be conducted. Retention of the alloy phase after electrocatalytic reactions will not be validated without deconvolution of XPS peaks.
13. ICP measurements of electrolyte after HER and HzOR half-cell reactions should be conducted to confirm any catalyst leaching.
14. In Figure S9, the morphology of the catalyst after SEM and TEM is very different. Show low resolution TEM to corroborate these two.
15. The HER activities in neutral and alkaline sea water in Figure S14a (LSV) and b (Chronoamperometry) do not match. In Fig. S14b, at 100 and 400 mV overpotential, current densities should be more if matched with S14a.
16. Perform all the three electrode measurements in alkaline medium using Hg/HgO/OH⁻ as reference electrode, since Ag/AgCl is not stable in alkaline medium.
17. What is the individual role of Ni and Co in hydrazine oxidation and HER? The activities should be checked with Ni@C/MXene and Co@C/MXene. What is the origin of F on MXene surface?
18. Provide reproducibility plots for three-electrode based HER, HzOR and two-electrode based HER coupled HzOR.

We thank the editor and the reviewers for their time and very valuable comments in improving the quality of this manuscript. Provided below is our detailed response to each question.

Reviewer 1

High energy consumption and chlorine evolution/corrosion are two critical obstacles to the practical deployment of seawater electrolysis technology for a long period. This work by Qiu and co-workers offers a rather good opportunity to breaking both bottlenecks. New chemistry of hybrid seawater splitting enables impressive performance in reducing electricity consumption and material cost for affordable hydrogen production while treating hydrazine pollution effectively at industrially large current. Successful demonstration of self-powered electrolyzers further indicates the potential of this technology for sustainable hydrogen production by renewables-driven seawater electrolysis. Systematic experiments and theoretical calculations have been conducted to characterize the catalyst and catalytic mechanism in a well-organized form. I would like to recommend the acceptance of this work with a minor revision according to the following points:

Our Response: We are very grateful to the reviewer for the encouraging and constructive comments.

Q1. The authors should compare the HER activity of the present catalyst with commercial Pt/C under similar conditions. The latter is a standard catalyst for HER.

Our Response: We thank the reviewer for the good suggestion. We have compared the HER activity of our catalyst with commercial 20% Pt/C in 1.0 M KOH, alkaline or neutral seawater. Overall, it shows comparable performance with Pt/C under similar conditions. This catalyst could even outperform the Pt/C at the current densities above 120 mA cm⁻² for all cases, making it attractive for large-current hydrogen production. Related data has been added as Supplementary Figure 19a, b in the revised Supporting Information. Necessary discussion was also added in the section of “Half-cell HER performance of NiCo/MXene-based electrode”, page 9 in the revised MS, highlighted in yellow.

Figure 1. LSVs of NiCo@C/MXene/CF and 20 wt.% Pt loaded on CF (Pt/CF) for HER in (a) 1.0 M KOH and (b) neutral or alkaline seawater with 1.0 M KOH.

Q2. Whether the reference electrodes were calibrated against RHE? Necessary correction should be conducted if not.

Our Response: We have calibrated the reference electrode against RHE for all the tests. The reference electrode was calibrated in high purity H₂ saturated electrolyte with Pt as the working electrode. The cyclic voltammograms (CVs) were measured at a scan rate of 1.0 mV s⁻¹, and the average of the two potentials at which the current crossed zero was taken as the thermodynamic potential (vs. Ag/AgCl electrode) for the hydrogen electrode reactions. The calibration method has been added as a section of “Calibration of the reference electrode” in the revised Supporting Information, page 2, highlighted in yellow.

Q3. The calculation was performed for a pathway of N₂H₄ → N₂H₄* → N₂H₃* → N₂H₂* → N₂H* → N₂*. Are there any other possible reaction ways especially those involving NH₃? The authors are suggested to provide related discussions with supporting calculation results.

Our Response: We appreciate the reviewer for the excellent suggestion. Three pathways have been proposed for HzOR in alkaline medium (e.g., *J. Power Sources*, 2008, 182, 520):

Since high-purity N_2 was detected as the only anodic product by gas chromatography (GC), a $4e$ pathway (reaction 3) should be responsible for HzOR on our catalyst in alkaline medium. Reaction 1 and 2 can be ruled out since no NH_3 or H_2 was generated by HzOR. This finding is also consistent with the literature reports (*e.g.*, *Nat. Commun.*, 2018, 9, 4365; *Nat. Commun.*, 2019, 10, 4514; *Sci. Adv.*, 2020, 6, eabb4197; *Chem. Sci.*, 2019, 10, 378; *Angew. Chem. Int. Ed.*, 2021, 60, 5984). Therefore, a $4e$ pathway (reaction 3) was reasonably applied for the theoretical calculation to gain an atomic insight into the catalytic mechanism of HzOR in our work. Related discussion has been added in the section of “Origin of the intrinsic activity of NiCo sites for promoting HzOR”, page 14 in the revised MS, highlighted in yellow. To clarify the anodic and cathodic product, the original GC spectra are split into two figures in Supplementary Figure 24 in the revised Supporting Information.

Figure 2. The signals of anodic gas product detected by gas chromatography.

Q4. In Fig. 4c, what is the meaning of ASE and OSS, as well as HSE and HSS? The authors should clearly indicate these abbreviations.

Our Response: We sincerely apologize for this typo error. The abbreviations of ASE and HSE refer to the alkaline seawater electrolyzer and the hybrid seawater electrolyzer, respectively. The MS has been carefully and thoroughly polished to eliminate all the possible mistakes.

Q5. A very recent benchmark work (*Angew. Chem. Int. Ed.* 2021, 60, 5984) on electrocatalytic hydrazine oxidation can be cited to enrich the background of this work.

Our Response: We would like to thank the reviewer for bringing this excellent and highly related work to our attention. It has been added as Ref. 28 in the revised MS, highlighted in yellow.

Q6. A typo error, the “Mm” in the y-axis in Fig. 4d, needs to be corrected.

Our Response: We sincerely apologize for this typo error. It has been corrected in Fig. 4d in the revised MS. We have carefully revised the MS to eliminate all the possible mistakes.

Q7. In Fig. 4g, the hydrogen yielding rate kept stable but hydrazine concentration was rapidly reduced in the same electrolyzer. Can the author explain it?

Our Response: We thank the reviewer to point this out. The hydrogen yielding rate is primarily determined by the current density instead of hydrazine concentration. It would keep stable when a constant current density is applied. The reduction of hydrazine concentration mainly raises the cell voltage to increase energy consumption.

Reviewer 2

Overall: This is a very interesting paper showing thermodynamically favorable hydrazine oxidation to address two extreme challenges of seawater electrolysis: the huge electricity consumption and anode corrosion by chlorine chemistry. They have shown high a faradaic current efficiency with less power consumption. The idea of using hydrazine oxidation rather than water is novel and the idea can be used in other areas. The article can be published after addressing the following comments:

Our Response: We are very grateful to the reviewer for the encouraging comments.

In “Introduction”

Q1. “Efficient technologies are thus highly desired to degrade the hydrazine in surface water to a trace residual” (page 4). What is the typical hydrazine concentration in wastewater and are they enough to produce energy for practical applications? If hydrazine is added then the authors need to do cost analysis to show the actual cost-effectiveness of the hybrid seawater electrolyzer system over conventional ones as they have used hydrazine, a value added product.

Our Response: We thank the reviewer for the excellent suggestion. The concentration of hydrazine in wastewater is highly varied in different industrial applications. For example, the chemical process involving hydrazinolysis reaction can yield wastewater with above 5 – 10 % hydrazine residual. It is far higher than the level of hydrazine concentration in our work (*ca.* 1.5 %), which is sufficient for energy-saving hydrogen production.

Use of commercial hydrazine leads to a hydrogen cost of *ca.* 2.38 USD per m³ H₂ assuming an average electricity cost of 0.11 USD per kWh in China and hydrazine (80 %) cost of 2,415 USD per ton (price at Apr 2020). Electrolysis of costless hydrazine sewage can largely reduce this cost to below 0.37 – 0.42 USD. It far excels existing water and seawater electrolyzers suffering the additional cost of seawater desalination and frequent anode maintenance besides unaffordable electricity consumption. The expense of our technique is anticipated to be even reduced by coupling renewable solar or wind power into the system. Meanwhile, the extra function of our hybrid electrolyzer in toxic hydrazine treatment further adds inestimable benefits to the protection of the ecosystem and human health. This merit is hardly realized by conventional ones with

less environmental sustainability. Related discussion has been added as a section of “Cost analysis of hybrid seawater electrolysis” in the revised Supporting Information, page 2 – 3, highlighted in yellow.

Q2. Seawater and wastewater matrices are completely two different systems. This is unfeasible to use them together.

Our Response: We thank the reviewer to point this out. The seawater and wastewater are simultaneously involved in the same hybrid electrolyzer but play different roles in different parts of the electrolyzer. The seawater is used for hydrogen production in cathodic chamber, while the wastewater is treated in anodic chamber.

In Results

Q3. “While the MXene layer with abundant -OH, -O and -F groups may effectively attract the water and hydrazine molecules via hydrogen bonding”. -F groups don’t do hydrogen bonding effectively.

Our Response: We agree with the reviewer that -F group doesn’t do hydrogen bonding effectively. Related description has been removed from the revised MS.

In “Half-cell HzOR performance of NiCo/MXene-based electrode”

Q4. “The HzOR activity of NiCo@C/MXene/CF is evaluated in 1.0 M KOH with various hydrazine concentrations”. Why did the authors use 1.0 M KOH when their electrolyzer is based on seawater with neutral pH? Reaction mechanism depends on solution pH critically. The authors need to do all the experiments (OER+HER) in seawater pH mimicked electrolyte and calculate all the parameters.

Our Response: We thank the reviewer to point this out. In our work, all the HzOR in half cells and anodic chamber of hybrid seawater electrolyzer were conducted under identical pH conditions (1.0 M KOH with 0.5 M hydrazine). Seawater is only used for HER at the cathode side in our hybrid electrolyzer.

We fully agree with the reviewer that the reaction mechanism critically depends on solution pH. The HER activity of our catalyst has been evaluated in seawater pH

mimicked electrolyte (pH 8.3) in three-electrode cells. The performance of the hybrid electrolyzer was also investigated by using seawater pH mimicked catholyte. In both cases, the performance of our catalyst and electrolyzer is rather comparable with that using neutral seawater. The OER is not considered because it is not involved in our electrolyzer. Related data has been added as Supplementary Figure 20 in the revised Supporting Information. Necessary discussion was also added in the section of “Half-cell HER performance of NiCo/MXene-based electrode” on page 9 and “Performance of hybrid seawater electrolyzer” in the revised MS on page 10, highlighted in yellow.

Figure 1. (a) The LSVs of NiCo@C/MXene/CF for HER in neutral seawater and the seawater pH (8.3) mimicked electrolyte. (b) The LSVs of the hybrid electrolyzer using seawater pH (8.3) mimicked catholyte.

In “Half-cell HER performance of NiCo/MXene-based electrode”

Q5. The authors stated the usefulness of the use of MXene. However, they need to state at least some reasons or hypothesis for its better activity. Is it only conductivity?

Our Response: We appreciate the reviewer for the good suggestion. The benefit of MXene in enhancing catalyst performance has been described in the section of “Effect of interfacial properties on electrolysis performance” on page 14 – 16 in the original MS.

Besides superb conductivity, the MXene with rich -OH and -O groups on the surface strengthens the attraction of water and hydrazine molecules onto the electrocatalytic interface via hydrogen bonding. These improvements have been validated by contact angle tests and quantitatively measured by electrochemical quartz crystal microbalance (EQCM). Such highly hydrophilic and hydrazine-friendly properties are essential to

achieving high access of water or hydrazine molecules to the inner Helmholtz plane above the electrocatalytic interface, thereby enhancing the catalytic kinetics. Moreover, the MXene also guides the chemical coupling of superaerophobic NiCo@C nanoarray onto electrode surface for minimizing gas bubble shielding on the electrode-electrolyte interface. Overall, it plays multiple roles in optimizing interfacial conductivity and robustness, water/hydrazine adsorption and gas-releasing capability for boosting large-current electrolysis.

In “Performance of hybrid seawater electrolyzer”

Q6. “seawater as the catholyte and 1.0 M KOH with 0.5 M hydrazine as the anolyte feed.” As two different solution pH is on the both sides of the electrolyzer how does the pH gradient over the membrane affect the performance?

Our Response: We thank the reviewer for the valuable suggestion. We have measured the performance of the hybrid seawater electrolyzer using seawater with various OH^- concentrations (0.0 – 3.0 M) as the catholyte and 1.0 M KOH containing 0.5 M N_2H_4 as the anolyte. It may give some clues on the effect of pH gradient over the anion exchange membrane (AEM) on electrolyzer performance. The pH gradient across AEM would be firstly reduced until OH^- concentrations in the catholyte rising to the same with the anolyte (1.0 M). Meanwhile, the HER activity is improved with the catholyte pH increasing, leading to a fast increase in cell performance. In this case, the direction of pH gradient over AEM is opposite to that of OH^- diffusion, thereby playing no significant role in electrocatalytic enhancement. After the catholyte pH exceeds the anolyte value, the direction of pH gradient across AEM would be the same with OH^- diffusion. But the performance of the electrolyzer is not significantly enhanced at large current densities due to the limitation of AEM in ionic exchange capacity and permeability. Developing high-performance AEM is desired to address this issue for full exploitation of the potential of our hybrid electrolyzer design. Related data have been added as Supplementary Figure 26 in the revised Supporting Information with necessary discussion in the section “Performance of hybrid seawater electrolyzer” on page 11 – 12 in the revised MS, highlighted in yellow.

Figure 2. (a) The LSVs of hybrid seawater electrolyzer using seawater with various OH⁻ concentrations as the catholyte and 1.0 M KOH containing 0.5 M N₂H₄ as the anolyte; (b) The correlation of cell voltage with OH⁻ concentrations in the catholyte at 500 mA cm⁻².

Q7. The authors have missed couple of significant work by other authors:

Dresp, S., Thanh, T. N., Klingenhof, M., Brückner, S., Hauke, P., & Strasser, P. (2020). Efficient direct seawater electrolyzers using selective alkaline NiFe-LDH as OER catalyst in asymmetric electrolyte feeds. *Energy & Environmental Science*, 13(6), 1725-1729.

Gayen, P., Saha, S., & Ramani, V. (2020). Selective seawater splitting using pyrochlore electrocatalyst. *ACS Applied Energy Materials*, 3(4), 3978-3983.

Our Response: We would like to thank the reviewer for bringing these excellent and highly related works to our attention. They have been added as Ref. 12 and 13 in the revised MS, highlighted in yellow.

Reviewer 3

In this manuscript the authors coupled hydrazine oxidation at the anode with sea water splitting at the cathode. This yields hydrogen at a very low cell voltage which is impressive. They have used NiCo alloy on MXene coated Cu foam as a catalyst. The result is good mainly because thermodynamic potential of hydrazine oxidation (HzOR) is very low compared to water oxidation and not due to selectivity of the catalyst. Previously also urea oxidation instead of water oxidation is chosen as anodic reaction to lower the cell voltage. Moreover, the Cl₂ evolution from seawater is not prominent in alkaline medium according to the previous reports. So, here also catalyst is not playing any prominent role from the 'result' point-of-view. I could not find any novelty, conceptual advancements, rational and intellectual thinking in terms of choosing the material. This work is not suitable for publication in Nature communications. Following are the comments to improve quality of the work before submission elsewhere.

Our Response: We thank the reviewer for the recognition on the impressive performance of our technology. The novelty and significance of our work lie in the new chemistry of hybrid seawater splitting. This advance enables low-cost hydrogen production from seawater at dramatically low electricity expense while avoiding CO₂ emission from the reported process like urea oxidation, which has not been achieved.

For seawater electrolysis in alkaline medium, a major challenge is the generation of corrosive ClO⁻ that rapidly destroy the anode (*e.g.*, *Energy Environ. Sci.*, 2020, 13, 1725; *PNAS*, 2019, 116, 6624). General approaches using the protective layer or chlorine-free anolyte are hard to eliminate the chlorine crossover and corrosion for long-term seawater electrolysis. Our technology offers a new way to fully avoid this problem by reducing the anodic potential to far below the chlorine oxidation potential, enabling better sustainability and electrolysis efficiency. It determines the novelty of our work in addressing the major obstacle of seawater electrolysis.

We have witnessed the effort in reducing the cell voltage by replacing the oxidation of water with urea (*e.g.*, *Adv. Funct. Mater.*, 2020, 30, 2000556; *Appl. Catal. B*, 2021, 284, 119740; *Nano Energy*, 2019, 60, 894). However, urea oxidation has to overcome sluggish 6e-involved kinetics to release CO₂. The resultant high cell voltages (1.4 – 1.8 V) and carbon emission fundamentally limit the practical feasibility and environmental sustainability of this technology.

A feature of our hybrid seawater electrolysis is the high-volume gas release from both anodic HzOR and cathodic HER, which severely deteriorates the large-current electrolysis performance. We devoted to addressing this problem by reasonably designing the catalyst with superaerophobic-hydrophilic and hydrazine-friendly properties. It works effectively in achieving high electrode activity for seawater electrolysis at an industrial-scale current level. Meanwhile, this catalyst design also strengthens the water adsorption to propel the rate-limiting Volmer step of HER while enabling efficient hydrazine attraction to promote HzOR. An overall enhancement in these aspects contributes greatly to the impressive performance of our electrolyzer. It reflects our rational and intellectual thinking in catalyst design for satisfying the hybrid seawater splitting chemistry.

Q1. During synthesis of the final catalyst, why NH_3 is used for annealing? The annealing in presence of NH_3 can form NiCo nitride instead of the alloy. Moreover, XRD reflections of NiCo alloy and nitride appears at similar position. So why it is an alloy and not its nitride?

Our Response: We thank the reviewer to point this out. The NH_3 is used to etch the carbon covered on NiCo alloy upon annealing for exposing more active sites. Our experience shows that the carbon coverage greatly reduces the catalyst performance and should be removed.

Figure 1. The LSVs of the catalyst obtained by annealing in NH_3 or H_2/Ar for (a) HzOR in 1.0 M KOH with 0.5 M N_2H_4 and (b) HER in 1.0 M KOH.

The NiCo alloy shows distinct XRD diffraction patterns to nickel or cobalt nitride, which can be easily distinguished from each other. No signals of nickel or cobalt

nitrides are observed by XRD. Their formation is avoided by annealing the NiCo-MOF precursor at a relatively low temperature. We also conducted the XAFS analysis to rule out the formation of nickel or cobalt nitrides. The NiCo@C, Ni@C and Co@C nanosheets were peeled off from the CF to minimize the influence of CF on the analysis. The K-edge XANES spectra of Ni and Co in NiCo@C are close to that of Ni@C, Co@C and metal foil reference, suggesting a metallic state of these elements. Curve fitting of FT-EXAFS spectra reveals the change of coordination number of Ni from 8.3 in Ni@C to 9.6 in NiCo@C while the value of Co increases from 8.4 in Co@C to 9.0 in NiCo@C. This phenomenon indicates the formation of NiCo alloy instead of their mixture. The presence of Ni-Ni bonds in NiCo alloy is identified by a similar metal bond length in NiCo@C (2.64 Å) and Ni@C (2.63 Å). The alloying of Co with Ni with a smaller atomic size induces a shorter metal bond length of 2.56 Å with respect to Co-Co bonds in Co@C (2.63 Å). It implies the atomic dispersion of Co atoms in Ni lattice in NiCo alloy. Tiny oxidization states (Ni-O, Co-O) appear for all the samples and metal foil references due to inevitable surface oxidation during XAFS analysis in air. No signals of Ni-N or Co-N bonds are detected.

Related data have been added as Supplementary Figure 3b and Supplementary Figure Fig. 4 in the revised Supporting Information. Necessary discussion has been added in the section of “Synthesis and characterization of NiCo/MXene-based electrode” on page 6 in the revised MS, highlighted in yellow.

Figure 2. XRD patterns of NiCo@C/MXene peeled off from the CF. The standard patterns of Ni₃N, Co₄N, Ni and Co are also presented for comparison.

Figure 3. (a, c) The normalized Ni K-edge XANES and the corresponding k^2 -weighted FT-EXAFS spectra of NiCo@C, Ni@C and Ni foil. (b, d) The normalized Co K-edge XANES and the corresponding k^2 -weighted FT-EXAFS spectra of NiCo@C, Co@C and Co foil.

Q2. The role of MXene in enhancing the conductivity and ECSA is not clear. It is mentioned that MXene is more conducting than soft carbon. But NiCo alloy is generated from MOF, so it is already assisted with conducting graphitic carbon which is shown by Raman spectra. What is the need of MXene? Why MXene enhances ECSA, whereas it is catalytically inert? What is the activity of MXene on copper foam towards HzOR and HER?

Our Response: We thank the reviewer to point this out. Although NiCo alloy is embedded in graphitic carbon, the MXene is necessary to ensure the intimate electrical contact between NiCo@C nanosheets and the current collector (Cu foam). More important, the MXene guides the uniform growth of dense NiCo@C nanoarrays on MXene/CF. Such highly porous architectures facilitate the exposure of active surface area and catalytic sites for improving the ECSA. Without MXene, irregular and easily peeled structures are formed on Cu foam due to poor chemical interaction between them.

Such a structural degradation largely reduces the ECSA and interfacial conductivity, thereby limiting the catalytic activity.

The activity of MXene on copper foam (MXene/CF) for HzOR and HER has been shown in Fig. 3d and Fig. 3f in the original MS, respectively. This material is nearly inactive for both reactions.

Figure 4. SEM images of NiCo@C grown on (a) MXene/CF and (b) bare CF. Scale bar, (a) 3 μm; (b) 2 μm.

Q3. In Fig. 2d, is it a single nanosheet or aggregation of nanoparticles? Where are the lattice fringes of MXene? Is it amorphous?

Our Response: Fig. 2d shows a single NiCo@C nanosheet on which abundant NiCo nanoparticles are embedded within the carbon matrix. The sample was peeled off from the NiCo@C/MXene/CF electrode for TEM analysis, so there are no lattice fringes of MXene.

Q4. Perform AFM imaging of NiCo@C/MXene/CF, NiCo@C/CF and MXene/CF to show the thickness and lateral length of each nanosheet.

Our Response: We appreciate the reviewer for the good suggestion. The AFM has been measured for the nanosheets on NiCo@C/MXene/CF. They have a thickness below 50 nm. The AFM analysis is not applicable for NiCo@C/CF and MXene/CF since there are no nanosheets on them. Related data have been added as Supplementary Figure 2 in the revised Supporting Information. Necessary discussion was added in the section of “Synthesis and characterization of NiCo/MXene-based electrode”, page 5 in the revised MS, highlighted in yellow.

Figure 5. (a) AFM image of NiCo@C nanosheets peeled off from the NiCo@C/MXene/CF. Scale bar: 500 nm. (b) Thickness curves of NiCo@C nanosheets.

Q5. I will recommend EXAFS and XANES analyses of the final catalyst as well as Ni@C/MXene and Co@C/MXene to show the Ni-Co binding and zero oxidation state of the alloy and change of coordination on going from individual metals to the alloy phase.

Our Response: We thank the reviewer for the excellent suggestion. Coordination states of the metal atoms in these catalysts are detected by XAFS. The NiCo@C, Ni@C and Co@C nanosheets were peeled off from the CF to minimize the influence of CF on the analysis. The K-edge XANES spectra of Ni and Co in NiCo@C are close to that of Ni@C, Co@C and metal foil reference, suggesting a metallic state of these elements. Curve fitting of FT-EXAFS spectra reveals the change of coordination number of Ni from 8.3 in Ni@C to 9.6 in NiCo@C while the value of Co increases from 8.4 in Co@C to 9.0 in NiCo@C. This phenomenon indicates the formation of NiCo alloy instead of their mixture. The presence of Ni-Ni bonds in NiCo alloy is identified by a similar metal bond length in NiCo@C (2.64 Å) and Ni@C (2.63 Å). The alloying of Co with Ni with a smaller atomic size induces a shorter metal bond length of 2.56 Å with respect to Co-Co bonds in Co@C (2.63 Å). It implies the atomic dispersion of Co atoms in Ni lattice in NiCo alloy. Tiny oxidization states (Ni-O, Co-O) appear for all the samples and metal foil references due to inevitable surface oxidation during XAFS analysis in air. Related data have been added as Supplementary Figure 4 in the revised MS with necessary discussion in the section of “Synthesis and characterization of NiCo/MXene-based electrode” on page 6 in the revised MS, highlighted in yellow.

Figure 6. (a, c) The normalized Ni K-edge XANES and the corresponding k^2 -weighted FT-EXAFS spectra of NiCo@C, Ni@C and Ni foil. (b, d) The normalized Co K-edge XANES and the corresponding k^2 -weighted FT-EXAFS spectra of NiCo@C, Co@C and Co foil.

Q6. In Figure S2, the XRD reflections of Cu foam also appear at almost the same position, how the authors could distinguish the peaks from the substrate and the catalyst? Does peeling of the catalyst eliminate all Cu? XRD pattern of bare Cu foam and also NiCo@C/MXene on Cu foam is recommended. Then the XRD of NiCo@C/MXene on Cu foam should also show two consecutive peaks, one due to the catalyst, another due to Cu foam. Where is the XRD pattern of individual metallic Co and Ni in Ni@C/MXene and Co@C/MXene?

Our Response: No necessary to distinguish the XRD peaks from the substrate and the catalyst because the XRD pattern in Figure S2 was collected from the NiCo@C/MXene or NiCo@C peeled off from the Cu foam.

The XRD pattern of Ni@C/MXene and Co@C/MXene peeled off from Cu foam was added as Supplementary Figure 3c in the revised Supporting Information. It reveals the formation of *fcc* Ni in Ni@C/MXene and *fcc* Co in Co@C/MXene. A rather similar

diffraction pattern of NiCo@C with Ni@C/MXene and Co@C/MXene indicates the *fcc* structure of NiCo alloy. Necessary discussion was added in the section of “Synthesis and characterization of NiCo/MXene-based electrode” on page 6 in the revised MS, highlighted in yellow.

Figure 7. XRD patterns of Ni@C/MXene, Co@C/MXene and NiCo@C/MXene.

Q7. In Figure S3, why the maximum binding is for Ti^{3+} and not due to Ti-C? Ar sputtering might help in increasing the intensity of Ti-C binding.

Our Response: We thank the reviewer to point this out. This maximum binding is not possible from Ti-C because its position is far from that of Ti-C and Ti^{2+} peaks of $Ti_3C_2T_x$ MXene. Strong Ti^{3+} signal may be a result of the reaction between oxygen-containing groups on MXene surface and Ti atoms connected to them upon annealing at high temperature (*e.g.*, $Ti_3C_2-OH \rightarrow Ti_3C_2-O$), which is common for MXene. After Ar^+ sputtering for 40 s, the intensity of Ti-C and Ti^{2+} peaks is increased noticeably, implying the Ti^{3+} species are mainly on MXene surface. Related data have been added as Supplementary Figure 5e in the revised Supporting Information. Necessary discussion was also added in the section of “Synthesis and characterization of NiCo/MXene-based electrode”, page 7 in the revised MS, highlighted in yellow.

Figure 8. Ti 2p XPS spectra of NiCo@C/MXene/CF after Ar⁺ sputtering.

Q8. Is Pt state-of-the-art catalyst for HzOR? Why the activities for this reaction compared with Pt?

Our Response: We compared our catalyst with Pt because the Pt represents a state-of-the-art catalyst and has been extensively used as a benchmark for evaluating the HzOR performance (*e.g.*, *Angew. Chem. Int. Ed.*, 2018, 57, 7649; *Nat. Commun.*, 2019, 10, 4514; *Angew. Chem. Int. Ed.*, 2016, 128, 703; *Adv. Energy Mater.*, 2019, 9, 1900390; *Nat. Commun.*, 2020, 11, 1853).

Q9. In Figure 3a, what is the reason for negative current below -0.1 V to -0.33V?

Our Response: The negative current below -0.1 to -0.33 V is due to simultaneous HER with a partially overlapped potential range. It is commonly observed in literature reports (*e.g.*, *Angew. Chem. Int. Ed.* 2018, 57, 7649; *Nat. Commun.* 2019, 10, 4514; *Sci. Adv.* 2020, 6, eabb4197).

Q10. Conducting LSV at 10 mV/s scan rate for any reaction exaggerates the activity. The authors should perform all the electrocatalytic measurements of the final catalyst at a lower scan rate like 1 mV/s.

Our Response: We thank the reviewer to point this out. To evaluate the effect of scan rate on catalyst activity, the LSVs of NiCo@C/MXene/CF were measured at a slow scan rate of 1.0 mV s⁻¹. It induces a rather slight improvement in HzOR and HER

activity in alkaline electrolytes possibly due to minimized double-layer charging. Overall, the variation of scan rates shows no significant effect on the electrochemical results. Related data has been added as Supplementary Figure 12 in the revised Supporting Information. Necessary discussion was also added in the section of “Half-cell HzOR performance of NiCo/MXene-based electrode” on page 8 and “Half-cell HER performance of NiCo/MXene-based electrode” on page 9 in the revised MS, highlighted in yellow.

Figure 9. A comparison of the LSVs of NiCo@C/MXene/CF for (a) HzOR in 1.0 M KOH with 0.5 M N₂H₄ and (b) HER in 1.0 M KOH at the scan rate of 1.0 and 10.0 mV s⁻¹.

Q11. What is the reason for noisy LSV plot for NiCo@C/CF?

Our Response: The noisy LSV plot of NiCo@C/CF is due to the notorious influence of gas bubbles released by HER and HzOR. Their continuous coverage and detachment repeatedly destroy and reconstruct the electrode-electrolyte interface. This effect leads to violent fluctuation of redox current and overpotential, especially at large current densities. This problem can be addressed by NiCo@C/MXene/CF electrode with superaerophobic properties for enhancing the performance of large-current electrolysis.

Q12. In Figure S9c and d, deconvolution of XPS peak of Ni and Co 2p should be conducted. Retention of the alloy phase after electrocatalytic reactions will not be validated without deconvolution of XPS peaks.

Our Response: The Ni 2p and Co 2p XPS spectra of NiCo@C/MXene/CF after HzOR have been deconvoluted in Supplementary Figure 14c, d in the revised Supporting Information. It suggests a negligible change of Co and Ni in metallic state after HzOR.

Q13. ICP measurements of electrolyte after HER and HzOR half-cell reactions should be conducted to confirm any catalyst leaching.

Our Response: We are grateful to the reviewer for the constructive suggestion. The composition of the electrolyte after HER and HzOR has been examined by ICP-OES. It reveals a very low residue of Ni, Co and Ti ions in the electrolyte, showing negligible catalyst leaching for long-term electrolysis. Related data has been added as Supplementary Table 2 in the revised Supporting Information. Necessary discussion was also added in the section of “Half-cell HzOR performance of NiCo/MXene-based electrode” on page 8 and “Half-cell HER performance of NiCo/MXene-based electrode” on page 9 in the revised MS, highlighted in yellow.

Table 1. The residue of Ni, Co and Ti ions in the electrolyte after long-term HzOR or HER in alkaline electrolytes.

Elements	HER	HzOR
	Concentration (mg L ⁻¹)	Concentration (mg L ⁻¹)
Ni	0.0005	0.0002
Co	0.0007	0.0004
Ti	0.001	0.0006

Q14. In Figure S9, the morphology of the catalyst after SEM and TEM is very different. Show low resolution TEM to corroborate these two.

Our Response: We thank the reviewer to point this out. We have provided the TEM image with a better resolution to validate the structure of the catalyst after HzOR tests. The size of NiCo@C nanosheets is too large to obtain the TEM image of entire sheet

with lower magnification. Related data has been added as Supplementary Figure 14b in the revised Supporting Information, highlighted in yellow.

Figure 10. (a) SEM of NiCo@NC/MXene/CF and (b) TEM image of NiCo@NC peeled from NiCo@NC/MXene/CF after accelerated durability test for HzOR in 1.0 M KOH with 0.5 M N₂H₄. Scale bar, (a) 2 μ m; (b) 50 nm.

Q15. The HER activities in neutral and alkaline sea water in Figure S14a (LSV) and b (Chronoamperometry) do not match. In Fig. S14b, at 100 and 400 mV overpotential, current densities should be more if matched with S14a.

Our Response: This variation is due to the different measuring protocols of LSVs and chronoamperometry. The LSVs are usually obtained with *iR* correction to evaluate the intrinsic activity of the catalyst. For chronoamperometry tests, the *iR* correction is not applied to reveal the practical performance of the catalyst. They are common protocols for HER electrocatalysis.

Q16. Perform all the three electrode measurements in alkaline medium using Hg/HgO/OH⁻ as reference electrode, since Ag/AgCl is not stable in alkaline medium.

Our Response: We agree with the reviewer's opinion on reference electrode stability and measure the HzOR and HER activity of our catalyst in alkaline medium against Hg/HgO reference electrode. For both reactions, the LSVs of NiCo@C/MXene/CF show a slight change relative to that measured against Ag/AgCl reference electrode. Good stability of our catalyst is also maintained for long-term HER and HzOR. Related data has been added as Supplementary Figure 13 in the revised Supporting Information.

Necessary discussion was also added in the section of “Half-cell HzOR performance of NiCo/MXene-based electrode” on page 8 and “Half-cell HER performance of NiCo/MXene-based electrode” on page 9 in the revised MS, highlighted in yellow.

Figure 11. A comparison of the catalytic performance of NiCo@C/MXene/CF, which are measured by using Ag/AgCl or Hg/HgO as the reference electrode. (a) LSVs and (b) chronopotentiometric curves at a current density of 100 mA cm⁻² for HzOR; (c) LSVs and (d) chronoamperometric curves for HER at $\eta = 100$ mV.

Q17. What is the individual role of Ni and Co in hydrazine oxidation and HER? The activities should be checked with Ni@C/MXene and Co@C/MXene. What is the origin of F on MXene surface?

Our Response: We are grateful to the reviewer for the good suggestion. The activity of Ni@C/MXene and Co@C/MXene has been evaluated for HzOR and HER. Both catalysts are active towards HzOR and HER in alkaline electrolytes. Specifically, the Co shows better HER activity than Ni, while the Ni is superior to Co for catalyzing HzOR. Their alloy enables an optimized bifunctionality for promoting HER and HzOR. The highest electrocatalytic activity is achieved by NiCo alloy with a Ni : Co ratio of 2.7 : 1.

Figure 12. LSVs of NiCo@C/MXene/CF, Ni@C/MXene/CF and Co@C/MXene/CF for (a) HER in 1.0 M KOH and (b) HzOR in 1.0 M KOH with 0.5 M N₂H₄.

Q18. Provide reproducibility plots for three-electrode based HER, HzOR and two-electrode based HER coupled HzOR.

Our Response: The reproducibility plots for three-electrode based HER and HzOR, and two-electrode based HER coupled HzOR have been added as Supplementary Figure 9 in the revised Supporting Information.

Figure 13. The reproducibility LSV plots of NiCo@C/MXene/CF electrode for (a) HzOR, (b) HER and (c) HER coupled HzOR for three tests.

REVIEWERS' COMMENTS

Reviewer #1 (Remarks to the Author):

In the revised version, the authors have carefully modified the manuscript by conducting additional experiments and providing related discussions&explanations, which have further enhance the quality of the manuscript. Therefore, the reviewer suggest the acceptance of the revised manuscript.

Reviewer #3 (Remarks to the Author):

I am quite satisfied with the revisions made by the authors. I will recommend the acceptance of this revised manuscript in its current form.

Reviewer 1

In the revised version, the authors have carefully modified the manuscript by conducting additional experiments and providing related discussions & explanations, which have further enhance the quality of the manuscript. Therefore, the reviewer suggest the acceptance of the revised manuscript.

Our Response: We highly appreciate the reviewer again for their valuable time and encouraging comments in improving the quality of this manuscript.

Reviewer 3

I am quite satisfied with the revisions made by the authors. I will recommend the acceptance of this revised manuscript in its current form.

Our Response: We are very grateful to the reviewer for their very constructive suggestions, which are truly helpful in enriching our work.